# MiniMax Learning of Interpretable Factored Stochastic Policies from Conjoint Data, with Uncertainty Quantification

## Abstract

We study offline learning of factored stochastic policies over extremely large, combinatorial action spaces and show how standard conjoint data can be used to estimate such policies with valid statistical uncertainty. Conjoint analyses typically report AMCEs by averaging over opponent attributes and thus ignore strategic interdependence. We instead learn *stochastic interventions*—product-of-Categorical policies over factor levels—that (i) optimize expected outcomes in an average-case setting and (ii) extend to a two-player *minimax* (adversarial) setting that realistically captures simultaneous strategic candidate selection. Methodologically, we derive a closed-form solution for the average-case optimizer under two-way interactions with $L_2$ variance regularization, and provide a general gradient-based procedure for richer model classes. Uncertainty from the outcome model propagates exactly to both the optimal policy and its value via the Delta method. We further model institutional details (e.g., primaries) inside the minimax objective and introduce a data-driven measure of strategic divergence between parties. On synthetic data, we characterize sample complexity and coverage as dimensionality and $n$ vary. On a U.S. presidential conjoint, adversarially learned policies produce equilibrium vote shares that align with historical election ranges, in stark contrast to non-adversarial (averaging) optimizers. To facilitate reproducibility and further research, we release an open-source dataset of mapped historical U.S. presidential candidate features on Hugging Face. Our framework connects causal policy learning with multi-agent RL in high-dimensional discrete action spaces while preserving interpretability and statistical guarantees.

Over the past decade, conjoint analysis, which is an application of high-dimensional factorial design, has become the most popular survey experiment methodology to study multidimensional preferences (Hainmueller et al., 2014). One of the most common political science applications of conjoint analysis is the evaluation of candidate profiles (e.g., Franchino & Zucchini, 2015; Ono & Burden, 2019; Christensen et al., 2021; Kirkland & Coppock, 2018).

In such experiments, respondents are asked to choose between two hypothetical political candidates whose features (e.g., gender, race, age, education, partisanship, and policy positions) are randomly selected. This design often employs a forced-choice format, where respondents must select one of the two candidates without an option to abstain or express no preference (Abramson et al., 2023). Researchers then proceed by estimating the average causal effect of each feature while marginalizing the remaining features over a particular distribution of choice. This popular quantity of interest is termed the Average Marginal Component Effect (AMCE) (Hainmueller et al., 2014).

AMCEs average a feature's effect over a chosen distribution for the other features, which means the answer depends on that averaging choice (de la Cuesta et al., 2019). In practice, researchers often use a uniform distribution, but real candidate pools are not uniform, and candidates do not choose their profiles in strategic isolation from opponents.

We therefore replace effect estimation with *policy learning*. Instead of asking for the marginal effect of a single attribute, we learn a *factored stochastic policy* over full profiles: a mixed distribution that assigns independent categorical probabilities to each attribute level and thus remains interpretable (one can read off how much weight the policy puts on, say, "economy" versus "immigration" as policy priority). In the non-adversarial, average-case setting, this policy is chosen to maximize expected win probability against a fixed reference

distribution for the opponent. We control variance and preserve interpretability by shrinking the learned policy toward the experimental assignment.

To capture strategic interaction, we extend to an adversarial setting in which both sides simultaneously choose their own mixed profile distributions. The objective is minimax—each side selects a profile distribution that is best against the other's—and institutional details are built in. Specifically, we model two stages: primary elections within each party (which induce a distribution over nominees) followed by the general election. The resulting equilibrium policies reflect how strategic opponents and institutions jointly shape feasible profile choices.

Under a linear outcome model with two-way interactions, the average-case optimizer with squared-distance regularization has a closed-form solution; uncertainty for both the optimal policy and its value is obtained by propagating the outcome-model uncertainty using the Delta method. For richer models, we optimize the logits of the factored policy by gradient methods while enforcing the simplex constraints implicitly, and we carry uncertainty through the optimization by differentiating end-to-end. The same machinery applies when we move from average-case to the adversarial, institution-aware game.

**Our contributions are:** (1) A shift from AMCEs to a conjoint estimand that is a factored stochastic policy over profiles, learned to maximize expected electoral performance, and interpretable at the attribute-level. (2) A closed-form average-case optimizer under two-way interactions with squared-distance variance control, together with uncertainty quantification for both the optimal policy and its value via the Delta method. (3) A general gradient-based procedure for richer outcome models (including regularized GLMs and neural models) with end-to-end differentiation for standard errors. (4) An adversarial, minimax extension that embeds institutional structure (primaries then general), yielding equilibrium mixed strategies and a data-driven measure of strategic divergence between parties. (5) Empirical evidence from simulations and a U.S. presidential conjoint showing that adversarially learned policies produce equilibrium vote shares aligned with historical ranges, plus an open-source mapping of historical candidate features to conjoint levels to facilitate replication.

**Related literature.** We contribute to the methodological literature on conjoint analysis and policy learning. In the growing field of conjoint analysis, we are, to our knowledge, the first to address optimal profile selection, particularly in adversarial settings. Related work in sequential decision-making includes multi-armed bandit problems (Audibert et al., 2010), while non-sequential conjoint studies focus on causal effect estimation (Hainmueller et al., 2014; Egami & Imai, 2019; de la Cuesta et al., 2019), hypothesis testing (Ham et al., 2022; Liu & Shiraito, 2023), causal estimand interpretation (Abramson et al., 2022), experimental design (Bansak et al., 2018), and stable preference analysis (Abramson et al., 2022; 2023). We also connect to causal inference literature on treatment rules, where recent advances in policy learning from granular data—both experimental and observational—have proliferated (see e.g., Dudik et al., 2011; Imai & Strauss, 2011; Zhao et al., 2012; Kitagawa & Tetenov, 2018; Athey & Wager, 2021; Ben-Michael et al., 2021; Kallus & Zhou, 2021; Zhang et al., 2022, and others), yielding individualized rules for binary treatments based on observables.

Our work can be seen as framing our problem as an offline contextual bandit with combinatorial actions, where conjoint randomization serves as the logging policy. The proposed optimal stochastic intervention learns a mixed policy over candidate attributes—a factorized Categorical distribution for interpretability and variance control—extending welfare-oriented policy learning (Kitagawa & Tetenov, 2018; Athey & Wager, 2021).The adversarial case models a two-player zero-sum game, with equilibria as minimax solutions to a single-step Markov game (Littman, 1994). Conjoint randomization identifies counterfactuals via direct modeling or off-policy estimators (e.g., doubly robust (Dudik et al., 2011)), sidestepping offline RL confounding (Levine et al., 2020). Overall, we adapt these tools to high-dimensional conjoints for interpretable stochastic policies and equilibria with uncertainty quantification.

## Learning Factored Stochastic Policies from Conjoint Data

Suppose that we have a simple random sample of $n$ respondents from a population. We consider a conjoint design of candidate choice with a total of $D$ factorial features per candidate. Each factorial feature $d \in \{1, 2, \ldots, D\}$ has $L_d \geq 2$ levels. The random variable representing an entire candidate profile presented to respondent $i$ in the design is labeled

$\mathbf{T}_i$. The support of $\mathbf{T}_i$, denoted $\mathcal{T}$, is the space of all possible treatment assignments and will vary based on the experimental design. For example, if each feature has $L$ levels, i.e., $L_1 = \cdots = L_D = L$, we have $\mathbf{t} \in \mathcal{T} = \{1, 2, ..., L\}^D$, where $\mathbf{t}$ is a specific realization of $\mathbf{T}_i$.

Usually, each respondent $i$ faces a choice between two candidate profiles, $\mathbf{T}_i^a$ and $\mathbf{T}_i^b$. The observed outcome will be an indicator of whether candidate $a$ is chosen over $b$, which occurs when the latent utility of $a$, represented as $Y_i(\mathbf{T}_i^a)$, is higher than that of $b$, which is represented as $Y_i(\mathbf{T}_i^b)$. This choice variable can be quantified as $C(\mathbf{T}_i^a, \mathbf{T}_i^b) = \mathbb{I}\{Y_i(\mathbf{T}_i^a) > Y_i(\mathbf{T}_i^b)\}$, representing the standard paired–profile conjoint in which respondents must select exactly one of two profiles as preferable, so that $C(\mathbf{T}_i^a, \mathbf{T}_i^b) \in \{0, 1\}$ (Hainmueller et al., 2014).

Often, each treatment combination is equally likely to be realized, in which case $\Pr(\mathbf{T}_i = \mathbf{t}) = |\mathcal{T}|^{-1}$ for all treatment combinations, $\mathbf{t}$. When factor levels have possibly different assignment probabilities, some treatment combinations will be more likely than others. Usually, each factor is assigned using draws from independent Categorical distributions so that we can write the probability of treatment combination $\mathbf{t}$ as

$$\Pr(\mathbf{T}_i = \mathbf{t}) = \prod_{d=1}^{D} \prod_{l=1}^{L_d} p_{dl}^{\mathbb{I}\{t_d = l\}},$$

where $p_{dl}$ is the Categorical probability for factor $d$ taking on level $l$ and $\mathbb{I}\{t_d = l\}$ is the indicator function that is 1 when $t_d$ takes on value $l$ and 0 otherwise. We let $\mathbf{p}$ define the vector of Categorical probabilities defining the data-generating distribution.

For simplicity, we make standard assumptions of conjoint analysis. That is, we assume that there is no interference between units and that the treatment assignment is randomized, i.e., $\{Y_i(\mathbf{t}^a), Y_i(\mathbf{t}^b)\}\{\mathbf{T}_i^a, \mathbf{T}_i^b\}$ and $\Pr(\mathbf{T}_i^c = \mathbf{t}^c) > 0$ for $c \in \{a, b\}$ and all $\mathbf{t}^c \in \mathcal{T}$.

**Optimal Selection of Conjoint Profiles in a Non-Adversarial Setting.** We consider the optimal selection of conjoint profiles, enabling us to study the types of political candidates who are likely to receive greater support from different types of voters. The standard approach, dominant in the policy learning literature, is to identify the following optimal treatment combination, $\mathbf{t}^* = \arg\max_{\mathbf{t} \in \mathcal{T}} \mathbb{E}[Y_i(\mathbf{t})]$, where $\mathbf{t}^*$ is the treatment combination that maximizes the average value of some generic outcome, $Y_i$. In the forced-choice conjoint case, this quantity would amount to $\mathbf{t}^{a^*} = \arg\max_{\mathbf{t}^a \in \mathcal{T}} \mathbb{E}[C(Y_i(\mathbf{t}^a), Y_i(\mathbf{T}_i^b))]$, so investigators find the vote-share-maximizing candidate profile $\mathbf{t}^{a^*}$, averaging over opposing candidate $b$ features (as in AMCE analysis). This approach has two limitations. First, high-dimensional treatments in conjoint analysis prevent identifying $\mathbf{t}^*$, as $|\mathcal{T}|$ far exceeds the sample size. Second, when multiple equally optimal profiles exist, identifying several is more informative than a single one.

To address this challenge, we propose finding an optimal stochastic intervention: we consider a parametric distribution of profiles $\Pr_{\boldsymbol{\pi}}(\cdot)$ that maximizes the average outcome. By considering a parametric model, we are able to effectively summarize a set of profiles that perform well. Formally, we seek the optimal stochastic intervention,

$$Q(\boldsymbol{\pi}^*) = \max_{\boldsymbol{\pi}} Q(\boldsymbol{\pi}) \quad \text{where} \quad Q(\boldsymbol{\pi}) = \sum_{\mathbf{t} \in \mathcal{T}} \mathbb{E}[Y_i(\mathbf{t})] \Pr_{\boldsymbol{\pi}}(\mathbf{T}_i = \mathbf{t}), \tag{1}$$

where $\boldsymbol{\pi}$ parameterizes the distribution of profiles. In the forced-choice conjoint case, this quantity can be written in the average case as agent $A$ optimizing their strategy, averaging over $B$'s fixed strategy:

$$Q(\boldsymbol{\pi}^{a^*}) = \max_{\boldsymbol{\pi}^a} \sum_{\mathbf{t}^a, \mathbf{t}^b \in \mathcal{T}} \mathbb{E}\left[C(Y_i(\mathbf{T}_i^a), Y_i(\mathbf{T}_i^b))\right] \Pr_{\boldsymbol{\pi}^a}(\mathbf{T}_i^a = \mathbf{t}^a) \Pr_{\mathbf{p}}(\mathbf{T}_i^b = \mathbf{t}^b).$$

The interpretation here is that $\boldsymbol{\pi}^{a^*}$ characterizes the highest possible vote share for a given counterfactual strategy of assigning the candidate characteristics of $a$, while features of $b$ are assigned according to a static averaging distribution (e.g., uniform).

Building on Equation 1, we preserve interpretability by restricting the counterfactual profile distribution for candidate $a$ to the same product-of-Categoricals used by the conjoint randomization $\Pr_{\mathbf{p}}$. Concretely, $\Pr_{\boldsymbol{\pi}^a}(\mathbf{T}_i^a = \mathbf{t}^a)$ has the identical factorized form

as $\Pr_{\mathbf{p}}(\mathbf{T}_i = \mathbf{t})$, but with per-attribute probabilities $\boldsymbol{\pi}^a$ replacing $\mathbf{p}$. This choice yields an attribute-readable policy and keeps off-policy evaluation tractable. The deterministic "best profile" appears as a degenerate special case, $\Pr_{\boldsymbol{\pi}^*}(\mathbf{T}_i = \mathbf{t}) = \mathbb{I}(\mathbf{t} = \mathbf{t}^*)$, but in high-dimensional conjoints that target is unidentifiable and statistically brittle; we therefore optimize over *stochastic* policies that summarize families of high-performing profiles.

To allow meaningful deviations from the design while controlling variance, we impose an $L_2$ (or KL) trust-region around the logging distribution:

$$\max_{\boldsymbol{\pi}^a} \ Q(\boldsymbol{\pi}^a) \ - \ \lambda_n \left\| \boldsymbol{\pi}^a - \mathbf{p} \right\|_2^2,$$

equivalently constraining $\|\boldsymbol{\pi}^a - \mathbf{p}\|_2 \le \epsilon_n$. This regularization is motivated by the increase in off-policy variance as $\boldsymbol{\pi}^a$ departs from $\mathbf{p}$. The restriction–regularization pair—matching the conjoint's factorized assignment and shrinking toward $\mathbf{p}$—is important because it (i) preserves interpretability, (ii) stabilizes estimation, and (iii) yields a closed-form average-case optimizer under two-way interactions (Proposition 1), while still admitting general gradient-based solutions for richer (e.g., neural) outcome models.

**Outcome model (Bernoulli GLM).** Let $C_i = \mathbb{I}\{Y_i(\mathbf{T}_i^a) > Y_i(\mathbf{T}_i^b)\}$ denote the forced choice. We model $C_i \mid (\mathbf{T}_i^a, \mathbf{T}_i^b) \sim \mathrm{Bernoulli}(\sigma(\eta_i))$, where $\sigma(x) = \{1 + \exp(-x)\}^{-1}$ is the logistic link and

$$\eta_i = \tilde{\mu} + \sum_{d=1}^{D} \sum_{l=1}^{L_d} \beta_{dl} \left( \mathbb{I}\{T_{id}^a = l\} - \mathbb{I}\{T_{id}^b = l\} \right) + \sum_{d<d'} \sum_{l=1}^{L_d} \sum_{l'=1}^{L_{d'}} \gamma_{dl,d'l'} \left( \mathbb{I}\{T_{id}^a = l, T_{id'}^a = l'\} - \mathbb{I}\{T_{id}^b = l, T_{id'}^b = l'\} \right).$$

(2)

where $\beta_{dl}$ denotes the main effect of factor $d$ with level $l$, $\gamma_{d'l',d''l''}$ denotes the interaction effect of treatment $d'l'$ and $d''l''$. We impose sum-to-0 constraints on $\{\beta_{dl}\}_l$ and on each $\{\gamma_{dl,d'l'}\}_{l,l'}$ for identifiability (Egami & Imai, 2019). Parameters $(\tilde{\mu}, \boldsymbol{\beta}, \boldsymbol{\gamma})$ are estimated via GLM (e.g., logistic) with optional sparsity (lasso) if desired. An intuition here is that the difference between utilities under candidates $a$ and $b$ defines the choice between $a$ and $b$.

This model makes calculation of the average outcome under a stochastic intervention straightforward if the policies, $\boldsymbol{\pi}^a$ and $\boldsymbol{\pi}^b$, over candidate $a$ and $b$ features define Categorical distributions. Then, the *Stochastic Intervention Under Forced Choice Conjoint* can be written as:

$$Q(\boldsymbol{\pi}^a, \boldsymbol{\pi}^b) = \mathbb{E}_{\mathbf{T}^a \sim \boldsymbol{\pi}^a, \, \mathbf{T}^b \sim \boldsymbol{\pi}^b} \left[ \sigma \left( \tilde{\mu} + \sum_{d,l} \beta_{dl} \left( \mathbb{I}\{T_d^a = l\} - \mathbb{I}\{T_d^b = l\} \right) + \sum_{d<d'} \sum_{l,l'} \gamma_{dl,d'l'} \left( \mathbb{I}\{T_d^a = l, T_{d'}^a = l'\} - \mathbb{I}\{T_d^b = l, T_{d'}^b = l'\} \right) \right) \right].$$

Under a linear probability approximation, this becomes:

$$Q(\boldsymbol{\pi}^a, \boldsymbol{\pi}^b) = \mathbb{E}_{\boldsymbol{\pi}^a, \boldsymbol{\pi}^b} \left[ \Pr(Y_i(\mathbf{T}_i^a) > Y_i(\mathbf{T}_i^b)) \right] = \tilde{\mu} + \sum_{d=1}^{D} \sum_{l=1}^{L_d} \beta_{dl} \left( \pi_{dl}^a - \pi_{dl}^b \right) + \sum_{d,d':d<d'} \sum_{l=1}^{L_d} \sum_{l'=1}^{L_{d'}} \gamma_{dl,d'l'} \left( \pi_{dl}^a \pi_{dl'}^a - \pi_{dl}^b \pi_{dl'}^b \right).$$

Motivated by the opponent candidate marginalization in AMCE analysis, we first consider the optimal average-case stochastic intervention where $a$ optimizes against a uniform distribution over candidate features. In this case, $\boldsymbol{\pi}^{a^*}$, can here be derived in closed form, assuming the features of the opposing candidate, $b$, are assigned according to a fixed distribution such as $\mathbf{p}$. We will call this kind of analysis the *Average Case Optimal Stochastic Intervention for Forced-Choice Conjoints* in that the behavior of the opponent, $b$, is static.

PROPOSITION 1 *Under a linear-probability approximation and with two-way interactions and $L_d$ levels for factor $d$, the average-case optimal $L_2$ regularized stochastic intervention is, for large enough value of $\lambda_n$, given by*

$$\boldsymbol{\pi}^{a^*} = \mathbf{C}^{-1} \mathbf{B}, \quad \text{where} \quad B_{r(dl),1} = -\beta_{dl} - 4\lambda_n p_{dl} - 2\lambda_n \sum_{l' \neq l, l' < L_d} p_{dl'}$$

$$C_{r(dl),r(dl)} = -4\lambda_n; \quad C_{r(dl),r(dl')} = -2\lambda_n; \quad C_{r(dl),r(d'l'')} = \gamma_{dl,d'l''},$$

*where $r(dl)$ denotes an indexing function returning the position associated with its factor $d$ and level $l$ into the rows of $B$ and rows or columns of $C$. For proof, see §A.I.2.*

Here, the optimal stochastic intervention, $\mathrm{Pr}_{\boldsymbol{\pi}^{a^*}}$, is a deterministic function of the outcome model parameters. The parameters defining the outcome model, $\boldsymbol{\beta}$ and $\boldsymbol{\gamma}$, are not known *a priori*, but can be estimated via GLM, with uncertainties calculated using asymptotic SEs.

Intuitively, the analysis done here allows researchers to investigate the implications of models for candidate choice fit on the data. Instead of examining marginals via AMCE, they can examine joint effects by looking at the optimal behavior implied under their choice of model. Estimates of the optimal distribution over candidates are generated using uncertain model parameters; however, the Delta method enables the rigorous propagation of uncertainty.

Because the values, $\boldsymbol{\pi}^{a^*}$ defining $\mathrm{Pr}_{\boldsymbol{\pi}^{a^*}}$ are a deterministic function of modeling parameters, the variance-covariance matrix of $\{\widehat{Q}(\widehat{\boldsymbol{\pi}}^{a^*}), \widehat{\boldsymbol{\pi}}^{a^*}\}$ can be obtained via the Delta method:

$$\text{Var-Cov}(\{\widehat{Q}(\widehat{\boldsymbol{\pi}}^{a^*}), \widehat{\boldsymbol{\pi}}^{a^*}\}) = \mathbf{J}\,\widehat{\Sigma}\,\mathbf{J}',$$

where $\widehat{\Sigma}$ is the variance-covariance matrix from the modeling strategy for $Y_i$ using regression parameters $\{\boldsymbol{\beta}, \boldsymbol{\gamma}\}$ and $\mathbf{J}$ is the Jacobian of partial derivatives (e.g., of $\widehat{Q}(\widehat{\boldsymbol{\pi}}^{a^*})$ and $\widehat{\boldsymbol{\pi}}^{a^*}$ w.r.t. the outcome model parameters): $\mathbf{J} = \boldsymbol{\nabla}_{\{\hat{\boldsymbol{\beta}}, \hat{\boldsymbol{\gamma}}\}}\{\widehat{Q}(\widehat{\boldsymbol{\pi}}^{a^*}), \widehat{\boldsymbol{\pi}}^{a^*}\}$. Under i.i.d., correct specification, regularity conditions, and standard moment conditions of the MLE,

$$\sqrt{n}\left(\{\widehat{Q}(\widehat{\boldsymbol{\pi}}^{a^*}),\ \widehat{\boldsymbol{\pi}}^{a^*}\} - \{Q(\boldsymbol{\pi}^{a^*}),\ \boldsymbol{\pi}^{a^*}\}\right) \to \mathcal{N}\left(\mathbf{0},\ \mathbf{J}\,\Sigma\,\mathbf{J}'\right).$$

The approach here thereby gives researchers a recipe for finding optimal stochastic interventions given choice of outcome model. Uncertainties from the outcome model parameters propagate into uncertainties over the optimal strategy. In sum, a closed-form expression for the regularized optimal stochastic intervention can be found in the base case of conjoint analysis where one candidate optimizes against a fixed opponent distribution.

**General Optimal Stochastic Interventions in Non-Adversarial Environments.** There are limitations to the approach just described. One limitation is that while preserving the sum-to-1 constraint on the probabilities, the analytical solution in Proposition 1 does not guarantee the non-negativity of $\hat{\boldsymbol{\pi}}^{a^*}$ for small values of $\lambda_n$. Another limitation is that, as soon as we generalize the outcome model to the GLM or $> 2$-way interactions, we have no analytical formula for the optimal solution.

To address these limitations, we can perform the stochastic intervention optimization for $\widehat{\boldsymbol{\pi}}^{a^*}$ using iterative methods instead of an analytical closed form.[1] For example, to ensure that the entries in $\hat{\boldsymbol{\pi}}^{a^*}$ lie on the simplex, we can re-parameterize the objective function using $\alpha_{dl}$'s, which inhabit an unconstrained space (see §A.I.7) for details). In particular, the stochastic interventional factor probabilities, $\boldsymbol{\pi}$, are now a function of $\boldsymbol{a}$, defined as so:

$$\mathrm{Pr}_{\boldsymbol{\pi}(\boldsymbol{a})}(T_d = l) = \begin{cases} \frac{\exp(\alpha_{dl})}{1 + \sum_{l'=1}^{L_d - 1} \exp(\alpha_{dl'})} & \text{if } l < L_d \\ \frac{1}{1 + \sum_{l'=1}^{L_d - 1} \exp(\alpha_{dl'})} & \text{if } l = L_d \text{ (baseline category)} \end{cases}$$

We can optimize this via gradient ascent, which, for almost every starting point, arrives at least at a local maximum or stationary point in polynomial time, assuming the strict saddle property of the function to be optimized (Lee et al., 2016). We update the unconstrained parameters using the gradient information, $\nabla_{\boldsymbol{\alpha}}\{O(\boldsymbol{a})\}$, where the full expression is found in §A.I.4. In particular, we iteratively update the initial state of $\boldsymbol{a}$ in $S$ gradient ascent update steps:

$$\underbrace{\left\{\text{for } s \in \{1, ..., S\}:\ \{\ \boldsymbol{\alpha}^{(s+1)} := \boldsymbol{\alpha}^{(s)} + \gamma^{(s)} \nabla_{\boldsymbol{a}}\{O(\boldsymbol{\alpha}^{(s)})\}\ \}\ \right\}}_{\text{gradients traced through all } S \text{ updates for Jacobian } \mathbf{J}}$$

Inference proceeds analogously to the closed-form case, via the delta method. With a closed-form expression for $\hat{\boldsymbol{\pi}}^{a^*}$, it is evident how we could write an expression for the derivative of optimal as a function of the regression parameters using the closed-form Jacobian. With an

---

[1]With two-way interactions, this setup can be framed as a quadratic programming problem with linear constraints, which can be solved efficiently using interior point or simplex methods. We focus on gradient ascent as a more general solution, allowing neural network models of outcome.

iterative computation needed to obtain $\hat{\boldsymbol{\pi}}^{a^*}$, we can consider the same quantity: although the closed-form derivatives of the iterative solution may be unknown, we can still evaluate these values using automatic differentiation—tracing the gradient information through the entire sequence of $S$ gradient ascent updates. Specifically, since the ascent procedure defines a deterministic mapping from $\hat{\boldsymbol{\beta}}, \hat{\boldsymbol{\gamma}}$ to $\hat{\boldsymbol{\pi}}^{a^*}$ (and thence to $\widehat{Q}(\hat{\boldsymbol{\pi}}^{a^*})$), reverse-mode differentiation can backpropagate sensitivities through the full unrolled sequence of $S$ updates, yielding $\mathbf{J}$.

**Adversarial Dynamics.** Thus far, we have considered optimal stochastic interventions under the assumption that one party (or candidate) chooses its profile distribution to maximize expected vote share, while treating the distribution of the opposing candidate's profile as fixed. Although this framework is useful in settings without direct strategic interaction (e.g., analyzing hiring choices), it is less suitable when two agents strategically select their own profiles in direct electoral competition. In many contexts, both the focal candidate and the opposing candidate are engaged in simultaneous strategic optimization.

To capture these adversarial dynamics, we introduce an *Adversarial Case Optimal Stochastic Intervention* framework that explicitly models two agents, which we label as $A$ and $B$, each attempting to maximize their expected probability of victory in a forced-choice setting. This is a two-player, simultaneous action zero-sum game. Let $Y_i(\mathbf{T}_i^c)$ represent respondent $i$'s latent utility for candidate $c \in \{A, B\}$, where $\mathbf{T}_i^c$ is the candidate's profile randomly drawn from some distribution. The observed forced-choice outcome is:

$$C(\mathbf{T}_i^A, \mathbf{T}_i^B) = \mathbb{I}\{Y_i(\mathbf{T}_i^A) > Y_i(\mathbf{T}_i^B)\}.$$

We define candidate profile distributions for $A$ and $B$ as $\boldsymbol{\pi}^A$ and $\boldsymbol{\pi}^B$, respectively. Each distribution assigns probabilities to the set of all possible profiles, $\mathcal{T}$. The choice of a stochastic (mixed) rather than deterministic profile stems from the combinatorics of potential profiles and impossibility of identifying a unique optimal profile with finite samples.

We consider a *zero-sum* environment where one candidate's gain is the other's loss. In this setting, it is natural to characterize the optimal profile distributions through a min-max optimization problem. Letting $Q(\boldsymbol{\pi}^A, \boldsymbol{\pi}^B) = \mathbb{E}_{\boldsymbol{\pi}^A, \boldsymbol{\pi}^B}\big[C(\mathbf{T}_i^A, \mathbf{T}_i^B)\big]$ denote the expected probability that candidate $A$ wins against candidate $B$, the adversarial objective is:

$$\max_{\boldsymbol{\pi}^A} \min_{\boldsymbol{\pi}^B} Q(\boldsymbol{\pi}^A, \boldsymbol{\pi}^B). \tag{3}$$

In equilibrium, neither candidate can improve their expected performance by unilaterally changing their distribution. Such a pair $(\boldsymbol{\pi}^{A^*}, \boldsymbol{\pi}^{B^*})$ constitutes a Nash equilibrium for the adversarial environment, evoking classic results in game theory (Kreps, 1989). In other words, given $\boldsymbol{\pi}^{B^*}$, no deviation from $\boldsymbol{\pi}^{A^*}$ improves $A$'s performance, and vice versa.

**Institutional Constraints.** Without institutional asymmetries, the adversarial game just described can admit trivial or symmetric equilibria. Real strategic environments such as elections, however, are structured by rules that determine *who* votes *when*, and therefore shape both feasible strategies and equilibria. We model a two-stage system—party primaries followed by a general election—under potentially asymmetric institutions across parties.

Let $A$ and $B$ index the two parties. Denote by $\mathcal{I}^A$ and $\mathcal{I}^B$ the (possibly overlapping) primary electorates for $A$ and $B$, respectively (closed, semi-open, and open primaries are special cases), and by $\mathcal{E}$ the general-election electorate. Institutions determine these sets and their sampling weights (e.g., turnout, inclusion of independents); we bundle these parameters as $\beth$. Each party $c \in \{A, B\}$ chooses a factored, product-of-Categorical mixed profile distribution $\boldsymbol{\pi}^c$ (our policy); the rest of that party's field is summarized by a counter-distribution $\boldsymbol{\pi}^{c'}$.[2]

**Primary Stage (Pairwise Head-to-head Model).** For two profiles $\mathbf{t}, \mathbf{t}' \in \mathcal{T}$ and party $c$, define the primary head-to-head win probability

$$\kappa_c(\mathbf{t}, \mathbf{t}') = \mathbb{E}_{i \in \mathcal{I}^c}\big[C(\mathbf{t}, \mathbf{t}')\big] = \Pr_{i \in \mathcal{I}^c}\big(Y_i(\mathbf{t}) > Y_i(\mathbf{t}')\big),$$

---

[2]Empirically, $\boldsymbol{\pi}^{c'}$ can be an empirical mixture over other entrants, a calibrated baseline, or set equal to $\boldsymbol{\pi}^c$ under symmetry. Multi-candidate or multi-round procedures can be handled by replacing the pairwise mechanism below with the appropriate implied choice probabilities.

where $C(\cdot, \cdot)$ is the forced-choice indicator introduced earlier. In a large electorate, $\kappa_c(t, t')$ approximates the majority outcome. A simple and differentiable pushforward from *primary strategies* to the *nominee distribution* is obtained by drawing one entrant from the party strategy and one from the field (drawn from the counter distribution, $\boldsymbol{\pi}^{c'}$):

$$\bar{\boldsymbol{\pi}}^c(\mathbf{t}) = \underbrace{\boldsymbol{\pi}^c(\mathbf{t})\, \mathbb{E}_{\mathbf{t}' \sim \boldsymbol{\pi}^{c'}}\big[\kappa_c(\mathbf{t}, \mathbf{t}')\big]}_{\mathbf{t} \text{ drawn as party entrant and wins}} + \underbrace{\boldsymbol{\pi}^{c'}(\mathbf{t})\, \mathbb{E}_{\mathbf{t}' \sim \boldsymbol{\pi}^c}\big[1 - \kappa_c(\mathbf{t}', \mathbf{t})\big]}_{\mathbf{t} \text{ drawn from field and defeats the party entrant}} . \quad (4)$$

Thus $\bar{\boldsymbol{\pi}}^c$ is the induced distribution over the party's nominee after the primary.

**Independence Across Primaries.** Conditional on profiles and voter utilities, we assume the two primaries resolve independently,

$$\Big\{ \mathbb{I}[Y_i(\mathbf{t}) > Y_i(\mathbf{t}')] \Big\}_{i \in \mathcal{I}^A} \perp \Big\{ \mathbb{I}[Y_j(\mathbf{u}) > Y_j(\mathbf{u}')] \Big\}_{j \in \mathcal{I}^B} \,\Big|\, \mathbf{t}, \mathbf{t}', \mathbf{u}, \mathbf{u}', \beth \quad (5)$$

which permits factorization of the joint nominee distribution as $\bar{\boldsymbol{\pi}}^A \otimes \bar{\boldsymbol{\pi}}^B$.

**General Election and Institutionalized Value.** Given nominees $\mathbf{t} \sim \bar{\boldsymbol{\pi}}^A$ and $\mathbf{u} \sim \bar{\boldsymbol{\pi}}^B$, the probability that $A$ wins the general election—averaging over the general electorate $\mathcal{E}$—is $\mathbb{E}_{i \in \mathcal{E}}\big[C(\mathbf{t}, \mathbf{u})\big] = \Pr_{i \in \mathcal{E}}\big(Y_i(\mathbf{t}) > Y_i(\mathbf{u})\big)$. The expected (institution-aware) payoff to $A$ is then

$$Q_{\text{inst}}\big(\boldsymbol{\pi}^A, \boldsymbol{\pi}^B;\, \boldsymbol{\pi}^{A'}, \boldsymbol{\pi}^{B'}, \beth\big) = \mathbb{E}_{\substack{\mathbf{t} \sim \bar{\boldsymbol{\pi}}^A(\boldsymbol{\pi}^A, \boldsymbol{\pi}^{A'}, \beth) \\ \mathbf{u} \sim \bar{\boldsymbol{\pi}}^B(\boldsymbol{\pi}^B, \boldsymbol{\pi}^{B'}, \beth)}} \Big[ \mathbb{E}_{i \in \mathcal{E}}\big[C(\mathbf{t}, \mathbf{u})\big] \Big] . \quad (6)$$

**Equilibria Under Institutions.** The minimimax problem over interpretable, variance-controlled policies using factored, product-of-Categorical distributions, $\Pi_{\text{fact}}^A, \Pi_{\text{fact}}^B \subset \Delta(\mathcal{T})$ becomes (assuming fixed $\boldsymbol{\pi}^{A'}, \boldsymbol{\pi}^{B'}$):

$$\max_{\boldsymbol{\pi}^A} \min_{\boldsymbol{\pi}^B} Q_{\text{inst}}\big(\boldsymbol{\pi}^A, \boldsymbol{\pi}^B;\, \boldsymbol{\pi}^{A'}, \boldsymbol{\pi}^{B'}, \beth\big),$$

defining a *restricted minimax* problem. Von Neumann's minimax theorem guarantees a saddle point on the full simplices $\Delta(\mathcal{T}) \times \Delta(\mathcal{T})$; however, within the restricted factored distribution class, $\Pi_{\text{fact}}^c$ is in general non-convex, so a saddle point need not exist within the restricted class. In practice, we compute a stationary point via gradient ascent–descent on unconstrained logits; see Appendix for a discussion of how to certify how close the learned pair $(\widehat{\boldsymbol{\pi}}^A, \widehat{\boldsymbol{\pi}}^B)$ is to a full mixed-strategy equilibrium.

We evaluate the pushforward map $\boldsymbol{\pi}^c \mapsto \bar{\boldsymbol{\pi}}^c$ by re-parameterized Monte Carlo over $\mathbf{t} \sim \boldsymbol{\pi}^c$ and $\mathbf{t}' \sim \boldsymbol{\pi}^{c'}$ and optimize via gradient-based ascent–descent on the unconstrained logits that parameterize the factored Categorical policies, with KL or $L_2$ variance-control regularization from Eq. 1. As in the non-institutional case, Delta-method inference follows by backpropagating sensitivities of $Q_{\text{inst}}$ to the outcome-model parameters through the unrolled optimization, yielding standard errors for $\widehat{\bar{\boldsymbol{\pi}}}^A, \widehat{\bar{\boldsymbol{\pi}}}^B$ and $\widehat{Q}_{\text{inst}}$.

**Remarks.** *(i)* Open vs. closed primaries, heterogeneous turnouts, and the participation of independents are encoded by $\beth$ via the composition/weighting of $\mathcal{I}^A$, $\mathcal{I}^B$, and $\mathcal{E}$. *(ii)* Hard rules (eligibility constraints, ballot-access requirements) are enforced by restricting the support of $\boldsymbol{\pi}^A$ and $\boldsymbol{\pi}^B$ to admissible profiles. *(iii)* Multi-round or multi-candidate primaries can be accommodated by replacing $\kappa_c$ with the appropriate implied choice probabilities; the pushforward Eq. 4 remains a differentiable functional of $(\boldsymbol{\pi}^c, \boldsymbol{\pi}^{c'}, \beth)$. The framework here can accommodate respondent covariates; see Appendix.

**Quantifying Strategic Divergence.** Unlike AMCE analysis, which cannot quantify observed candidate information through experimental findings, the methodology here enables the measurement of strategic divergence using actual candidate profiles and the elicited conjoint preferences. In particular, given the optimal candidate distribution for one party, $\boldsymbol{\pi}^A$, and another, $\boldsymbol{\pi}^B$, in a given institutional context, we can find the strategic divergence factor, $\mathcal{D}$, of a given candidate profile, $\mathbf{t}$, using the estimated strategies:

$$\mathcal{D}(\mathbf{t}) = \left| \log\left( \frac{\Pr_{\boldsymbol{\pi}^A}(\mathbf{t})}{\Pr_{\boldsymbol{\pi}^B}(\mathbf{t})} \right) \right| = \big| \log\{\Pr_{\boldsymbol{\pi}^A}(\mathbf{t})\} - \log\{\Pr_{\boldsymbol{\pi}^B}(\mathbf{t})\} \big|. \quad (7)$$

When $\mathcal{D}(\mathbf{t})$ is 0, the candidate profile $\mathbf{t}$ would be equally likely under the strategic action of party $A$ and $B$; when $\mathcal{D}(\mathbf{t})$ is large, this is an indication that a given profile would be likely under the strategy of one party, but unlikely under the strategy of another.

## EXPERIMENTS: SYNTHETIC SCALING & SAMPLE COMPLEXITY

**Average Case Simulation.** In Monte Carlo simulations using synthetic binary conjoint data under a linear outcome model with interactions (scaled to $R^2 = 0.70$ for main effects), we assess finite-sample convergence of the average-case optimal stochastic intervention by varying sample sizes ($n \in 500, 1500, 3500, 10000$) and dimensions ($K \in 5, 10, 20$), with $L_2$ regularization tuned to diverge moderately from the uniform data-generating distribution. Results demonstrate negligible bias and rapidly declining RMSE (variance-dominated) for $\hat{\boldsymbol{\pi}}^*$ and $\widehat{Q}(\hat{\boldsymbol{\pi}}^*)$ even at small $n$, with inference reliable as coverage nears nominal levels across settings; details, including Figures 4–9, are in §A.I.8. This approach also substantially outperforms a baseline AMCE-based policy (selecting per-factor maximizers), achieving higher expected outcomes on average; see Figure 5 and Appendix for details.

**Adversarial Case Simulation Design.** To assess finite-sample performance in the adversarial setting, we simulate two-party strategic competition between Republicans ($R$) and Democrats ($D$) in a two-stage electoral process: primaries for nominee selection, followed by a general election. Voters are affiliated with $R$ (fraction $p_R$) or $D$ ($p_D = 1 - p_R$), with only affiliated voters participating in their primary. We grid over $p_R \in \{0.2, 0.3, 0.5, 0.65, 0.8\}$ and sample sizes $n \in \{1000, 5000, 10000\}$, with Monte Carlo replications per cell.

In primaries, each party offers two profiles ($\mathbf{T}_i^{R,1}, \mathbf{T}_i^{R,2}$ for $R$; similarly for $D$), with one selected via the party's mechanism and the other uniform. Voter choices follow logistic models based on features (gender, for tractable ground-truth equilibria). General elections pit primary winners ($\mathbf{T}_i^{R,*}, \mathbf{T}_i^{D,*}$) against each other, with all voters choosing via separate $R$- and $D$-specific logistic models.

Ground-truth mixed strategies $\boldsymbol{\pi}^R, \boldsymbol{\pi}^D$ approximate Nash equilibria via grid search, maximizing $Q(\boldsymbol{\pi}^R, \boldsymbol{\pi}^D) = \mathbb{E}[\Pr(Y_i(\mathbf{T}_i^R) > Y_i(\mathbf{T}_i^D))]$. Equilibria satisfy $\max_{\boldsymbol{\pi}^R} \min_{\boldsymbol{\pi}^D} Q = \min_{\boldsymbol{\pi}^D} \max_{\boldsymbol{\pi}^R} Q$, where neither party can unilaterally improve expected vote share. For each run, we estimate equilibria and outcomes, evaluating how $p_R$ and $n$ affect strategies and vote shares. We report RMSE and 95% CI coverage for $\boldsymbol{\pi}^R$. (Details in §A.4.)

**Adversarial Case Simulation Results.** Simulation results indicate that the estimation error depends primarily on the conjoint sample size, with only modest sensitivity to the proportion of Republican voters. Larger sample sizes reduce uncertainty by stabilizing the estimates of voter utilities: with larger sample sizes, the overall estimation error declines sharply for all values of $p_R$. Coverage rates fall below the nominal level for $n = 1000$ but approach the nominal 95% level for larger sample sizes. The stronger performance under increasing $n$ reflects the fact that voters' utilities are more precisely estimated, allowing us to obtain better approximations of the zero-sum equilibrium in a two-party adversarial competition. In sum, these simulations highlight the key role of sample size and voter-party composition in estimating equilibrium strategies under adversarial conditions. Next, we apply these approaches to real data to explore optimal strategic dynamics in practice.

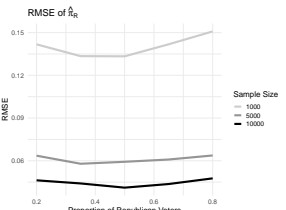 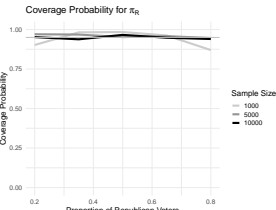

**Figure 1:** Finite-sample performance of $\widehat{\boldsymbol{\pi}}^R$ in the adversarial simulation. The top panel shows the root-mean-squared error (RMSE) of $\widehat{\boldsymbol{\pi}}^R$ for different sample sizes and proportions of Republican voters, $p_R$. The bottom panel illustrates the coverage probability of 95% confidence intervals for components of $\widehat{\boldsymbol{\pi}}^R$.

## EXPERIMENTS: REAL-WORLD CONJOINT (U.S. PRESIDENTIAL)

We now apply our methods to analyze policy positioning and optimal candidate selection using presidential preference data from Ono & Burden (2019). Here, our outcome is a

binary indicator stating whether candidate $a$ or $b$ was selected by respondent $i$ in a forced conjoint experiment. In the latent utility formulation above, this can be characterized as $C(Y_i(\mathbf{T}_i^a), Y_i(\mathbf{T}_i^b))$—an indicator indicating whether profile $\mathbf{T}_i^a$ yields higher utility for the respondent than profile $\mathbf{T}_i^b$; see §A.1 for a list of candidate factors. (Here, all standard errors for Delta method uncertainty propagation are clustered at the respondent level.)

**Average vs. Adversarial Case Results.** We optimize expected vote share for subpopulations (all, Democrats, Republicans, independents) against a uniform opponent distribution, using a GLM with interactions (lasso-regularized coefficients). Optimal stochastic interventions (Fig. 6) diverge on immigration, abortion, and policy expertise (e.g., economy vs. public safety preferences), but converge on personality traits. Under closed primaries, we compute minimax equilibria for Republican vs. Democrat strategies. Optimal policies (Fig. 7) differ from average-case (Fig. A.III.2); e.g., Democrats deprioritize immigration in average case but counter Republican guest-worker stances adversarially. Equilibrium vote shares drop markedly (Fig. A.III.1), aligning closer to historical elections.

**Results with Data-Driven Clustering.** Prior analyses of regularized optimal stochastic interventions—with and without adversarial dynamics—ignored respondent characteristics beyond party affiliation. Yet, heterogeneous voter types often favor distinct candidate profiles. To uncover these differences, we apply optimal stochastic interventions under data-driven respondent clustering, revealing how subgroups respond uniquely to high-dimensional features. Leveraging the clustered outcome model of Goplerud et al. (2022), Fig. 11 shows that covariate-sensitive strategies recover the underlying Democrat-Independent-Republican preference structure a priori, without explicit inputs. This highlights the approach's value in non-adversarial settings, where subgroup discovery enables tailored treatment strategies.

**Historical Comparison.** In contrast to AMCEs, our methods yield *distributions* over profiles, enabling likelihood-based evaluation of observed candidates. We map the 2016 primary contenders to the conjoint levels of Ono & Burden (2019) (details in §A.III.1; we make this historical data on observed candidate features available open source on Hugging Face (anonymous URL)); when a stance is ambiguous, we average uniformly over plausible levels. Fig. 2 shows that the average-case optimizer (uniform opponent) implies vote shares far outside the historical two-party range since 1976, whereas the adversarial optimizer closely matches the 2016 result and its confidence interval covers the historical range. We then score each 2016 contender by the log probability of their features under the estimated optimal stochastic interventions. As seen in the table accompanying Fig. 2, log probabilities under the adversarial strategies are higher than with average-case. Fig. 10 highlights heterogeneity in polarization. Finally, Fig. A.III.5 aggregates the strategic divergence factor from Eq. 7; overall, Democratic candidates show somewhat higher divergence.

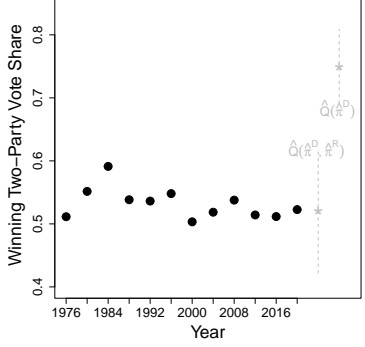

| Party | Quantity | Mean Log Prob. (s.e.) |
|---|---|---|
| Democrats | Average case | -16.18 (0.62) |
| Democrats | Adversarial case | -16.77 (0.71) |
| Democrats | Log likelihood ratio | -0.59 |
| Republicans | Average case | -15.87 (0.35) |
| Republicans | Adversarial case | -15.77 (0.37) |
| Republicans | Log likelihood ratio | 0.10 |

Figure 2: Comparing the average case and adversarial case results with real historical data. The adversarial case expected optimal outcomes are well within the range of historical experience; the average case outcomes are not.

LIMITATIONS. This approach has limitations (Table 1). Unlike non-parametric AMCE estimation, optimal stochastic interventions integrate all factors and strategic dynamics but rely on a two-step estimator requiring assumptions. While generalizable to complex outcome models (e.g., neural networks (Zhang et al., 2025)), inference is difficult without accessible variance-covariance matrices. Uncertainty estimates for equilibrium selection do not account for preference formation. And, inferred strategic behavior depends on institutional design, which may be hard to quantify. □

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

# Main Appendices

## A    Appendix

### A.1    Ono & Burden, Context

To fix ideas, we consider a conjoint analysis of candidate choice for the U.S. President, originally conducted by Ono & Burden (2019). The survey was fielded in March 2016, with choice of conjoint features influenced by the context of the 2016 US presidential elections. In this experiment, the outcome is a binary indicator stating whether one hypothetical candidate or the other was selected by a respondent. Candidate features were randomized with uniform probability and included age, sex, family context, and race.

The authors investigated various AMCEs, with a particular focus on gender. The AMCEs are computed non-parametrically by taking the difference between the fraction of female versus male candidates selected, averaging over all other (uniformly allocated) features of the candidate and its opponent. Fig. 3 summarizes all factor-level AMCEs; the authors concluded that female candidates are disadvantaged but the effect magnitude is small.

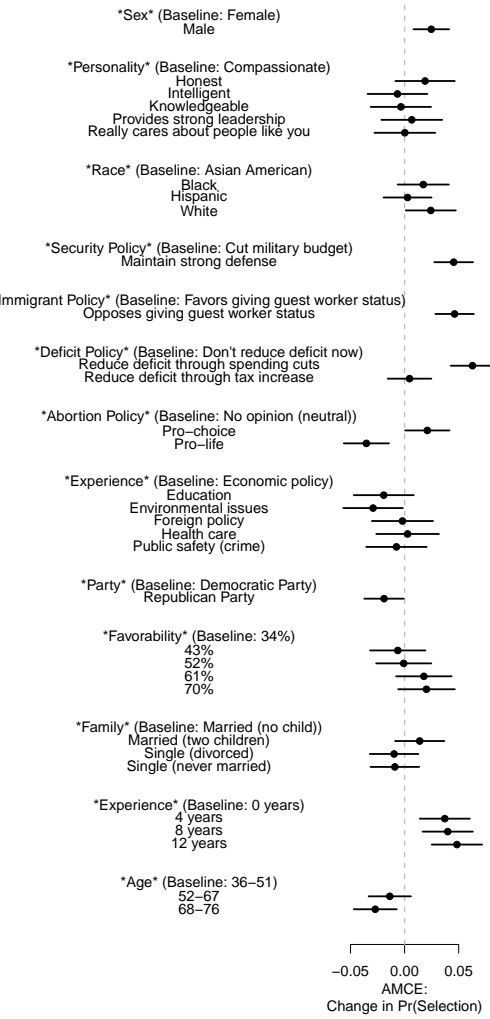

Figure 3: An AMCE analysis using presidential candidates from the conjoint data of Ono & Burden (2019).

### A.2 METHODOLOGICAL DETAILS

**Certifiability.**

DEFINITION 1 (RESTRICTED EQUILIBRIUM AND EXPLOITABILITY) *Let* $\Pi_{\text{fact}}^A, \Pi_{\text{fact}}^B$ *be the factored policy classes. A pair* $(\boldsymbol{\pi}^A, \boldsymbol{\pi}^B) \in \Pi_{\text{fact}}^A \times \Pi_{\text{fact}}^B$ *is a* restricted minimax equilibrium *if it is a saddle point of* $Q_{\text{inst}}$ *when each player is constrained to* $\Pi_{\text{fact}}^c$. *Given a candidate pair* $(\boldsymbol{\pi}^A, \boldsymbol{\pi}^B)$, *define*

$$\epsilon_{\text{ext}}^A \;=\; \max_{\sigma \in \Delta(\mathcal{T})} Q_{\text{inst}}(\sigma, \boldsymbol{\pi}^B;\, \boldsymbol{\pi}^{A'}, \boldsymbol{\pi}^{B'}, \sqsupset) \;-\; Q_{\text{inst}}(\boldsymbol{\pi}^A, \boldsymbol{\pi}^B;\, \boldsymbol{\pi}^{A'}, \boldsymbol{\pi}^{B'}, \sqsupset),$$

$$\epsilon_{\text{ext}}^B \;=\; Q_{\text{inst}}(\boldsymbol{\pi}^A, \boldsymbol{\pi}^B;\, \boldsymbol{\pi}^{A'}, \boldsymbol{\pi}^{B'}, \sqsupset) \;-\; \min_{\sigma \in \Delta(\mathcal{T})} Q_{\text{inst}}(\boldsymbol{\pi}^A, \sigma;\, \boldsymbol{\pi}^{A'}, \boldsymbol{\pi}^{B'}, \sqsupset).$$

*The* external exploitability *is* $\epsilon_{\text{ext}} = \max\{\epsilon_{\text{ext}}^A, \epsilon_{\text{ext}}^B\}$; $\epsilon_{\text{ext}} = 0$ *certifies a full mixed-strategy equilibrium.*

Next, we show that when the opponent's strategy is fixed, the institution-aware payoff is linear in the focal player's joint distribution, so the focal player's best response (over the full simplex) is always a pure profile and can be found by simply evaluating a scalar score for each profile. This lets us compute exploitability exactly (or by approximation) without solving an inner optimization.

PROPOSITION 2 (LINEARITY) *Fix* $\boldsymbol{\pi}^{A'}, \boldsymbol{\pi}^{B'}, \sqsupset$ *and* $\boldsymbol{\pi}^B$. *Then* $Q_{\text{inst}}(\boldsymbol{\pi}^A, \boldsymbol{\pi}^B)$ *is linear in* $\boldsymbol{\pi}^A$. *Specifically,*

$$Q_{\text{inst}}(\boldsymbol{\pi}^A, \boldsymbol{\pi}^B) \;=\; \sum_{\mathbf{t} \in \mathcal{T}} \boldsymbol{\pi}^A(\mathbf{t})\, G^A(\mathbf{t}),$$

*where* $V(\mathbf{t}, \mathbf{u})$ *denotes the general-election win probability for A when nominees* $\mathbf{t}$ *and* $\mathbf{u}$ *face off:*

$$V(\mathbf{t}, \mathbf{u}) \;:=\; \mathbb{E}_{i \in \mathcal{E}}\big[C(\mathbf{t}, \mathbf{u})\big] \;=\; \Pr_{i \in \mathcal{E}}\big(Y_i(\mathbf{t}) > Y_i(\mathbf{u})\big)$$

*and*

$$\Psi^B(\mathbf{t}) = \sum_{\mathbf{u}} \bar{\boldsymbol{\pi}}^B(\mathbf{u})\, V(\mathbf{t}, \mathbf{u}), \qquad G^A(\mathbf{t}) = \sum_{\mathbf{s}} \boldsymbol{\pi}^{A'}(\mathbf{s})\Big\{ \Psi^B(\mathbf{t})\, \kappa_A(\mathbf{t}, \mathbf{s}) + \Psi^B(\mathbf{s})\, [1 - \kappa_A(\mathbf{t}, \mathbf{s})] \Big\},$$

*as the expected general-election win probability for A if A nominates profile* $\mathbf{t}$ *and B's nominee is drawn from its post-primary distribution* $\bar{\boldsymbol{\pi}}^B$ *(which itself reflects the primary rules* $\sqsupset$*). Hence a full-simplex best response for A is the* pure *profile* $\mathbf{t}^\star \in \arg\max_{\mathbf{t}} G^A(\mathbf{t})$, *and* $\epsilon_{\text{ext}}^A = \max_{\mathbf{t}} G^A(\mathbf{t}) - \sum_{\mathbf{t}} \boldsymbol{\pi}^A(\mathbf{t}) G^A(\mathbf{t})$. *The symmetric statement holds for B.*

*Proof sketch.* Substitute the primary pushforward (Eq. 4) into $Q_{\text{inst}}$ and collect terms in $\boldsymbol{\pi}^A$.

**Covariate-Sensitive Strategies** The approach discussed here can accommodate respondent covariates, as is possible in the sequential decision-making context (Lu et al., 2010). In our discussion up to now, the new treatment probabilities were assigned without considering the specific characteristics of each respondent. We could consider stochastic interventions that took into account covariate information in the targeting of the high-dimensional treatments:

$$Q(\boldsymbol{\pi}^*) = \max_{\boldsymbol{\pi}} Q(\boldsymbol{\pi}) = \max_{\boldsymbol{\pi}} \mathbb{E}_{\mathbf{X}}\big[ \mathbb{E}_{Y|\mathbf{X}}\left[ Y_i(\mathbf{t}) \mid \mathbf{X}_i = \mathbf{x} \right] \Pr_{\boldsymbol{\pi}}\left( \mathbf{T}_i = \mathbf{t} \mid \mathbf{X}_i = \mathbf{x} \right) \big]$$

$$= \max_{\boldsymbol{\pi}} \left\{ \sum_{\mathbf{x}} \sum_{\mathbf{t} \in \mathcal{T}} \mathbb{E}[Y_i(\mathbf{t}) \mid \mathbf{X}_i = \mathbf{x}] \Pr_{\boldsymbol{\pi}}\left( \mathbf{T}_i = \mathbf{t} \mid \mathbf{X}_i = \mathbf{x} \right) \Pr(\mathbf{X}_i = \mathbf{x}) \right\}.$$

The covariate-sensitive distribution, $\Pr_{\boldsymbol{\pi}}(\mathbf{T}_i \mid \mathbf{X}_i)$, can be operationalized by having different factor-level probabilities for each cluster, with a model predicting the cluster probabilities for

each unit. If we let $\pi_{dlk}$ denote the probability of factor $d$, level $l$, for cluster $k \in \{1, ..., K\}$:

$$\Pr_{\boldsymbol{\pi}}(\mathbf{T}_i \,|\mathbf{X}_i) = \sum_{k=1}^{K} \Pr_{\boldsymbol{\pi}_k}(\mathbf{T}_i \,|Z_{ik} = 1) \, \Pr(Z_{ik} = 1|\mathbf{X}_i)$$

$$= \sum_{k=1}^{K} \underbrace{\left\{ \prod_{d=1}^{D} \Pr_{\boldsymbol{\pi}_k}(T_{id}|Z_{ik} = 1) \right\}}_{\text{Categorical probabilities for cluster } k} \, \underbrace{\Pr(Z_{ik} = 1|\mathbf{X}_i = \mathbf{x})}_{\text{Softmax regression}} \tag{8}$$

In this context, estimation can be conducted using outcome models that cluster main and interaction effects (Goplerud et al., 2022).

### A.3  SIMULATION DETAILS, AVERAGE CASE

**Simulation Design: Average Case.**  To probe finite-sample dynamics of the proposed optimal stochastic intervention methodologies for conjoint analysis, we employ Monte Carlo methods. In our simulations, we analyze synthetic factorial experiments with binary treatments where each treatment is drawn from an independent Bernoulli with probability parameter 0.5. We adopt a linear outcome model with interactions[3]:

$$Y_i(\mathbf{T}_i) = \beta_0 + \sum_{d=1}^{D} \sum_{l=1}^{L_d-1} \beta'_{dl} \mathbb{I}\{T_{id} = l\} + \sum_{d',d'':d'<d''} \sum_{l'=1}^{L_{d'}-1} \sum_{l''=1}^{L_{d''}-1} \gamma_{d',d''} \mathbb{I}\{T_{id'} = l'\}\mathbb{I}\{T_{id''} = l''\} + \epsilon_i,$$

with $\epsilon_i \sim N(0, 0.1)$, since this makes the computation of $Q(\boldsymbol{\theta})$ straightforward (in particular, $Q(\boldsymbol{\pi}) = \beta'\boldsymbol{\pi} + \sum_{d,d':d<d'} \gamma_{d,d'} \pi_d \pi_{d'}$). The coefficients are drawn i.i.d. from $N(0, 1)$, and the interaction coefficients are scaled so that the $R^2$ in using the main effects only to predict the outcome is 0.70 (ensuring some effective non-linearity). We obtain the true value of $\boldsymbol{\pi}^*$ fixing $\lambda$ and solving for $\boldsymbol{\pi}^*$ using Proposition 1.

To analyze finite sample convergence of $\widehat{\boldsymbol{\pi}}^*$, we vary the number of observations, $n \in \{500, 1500, 3500, 10000\}$. To analyze performance in the high-dimensional setting, where the number of treatment combinations is greater than the number of observations, we vary the number of factors, $K \in \{5, 10, 20\}$. We fix $\lambda$ so that the regularized optimal stochastic interventions have no factor probabilities greater than 0.9, while having a degree of divergence from the (uniform) data-generating probabilities.

**Simulation Results: Average Case.**  First, we examine the degree to which $\widehat{\boldsymbol{\pi}}^*$, the optimal stochastic intervention factor probabilities, and $\widehat{Q}(\widehat{\boldsymbol{\pi}}^*)$, the average outcome under the regularized stochastic intervention, converge to the true values as the sample size grows. We see in the left panel of Fig. 4 that, with a small number of factors (5), the bias of $\widehat{\boldsymbol{\pi}}^*$ is insignificant even with a small sample size (500). The variance of estimation contributes more prominently to the overall RMSE for all numbers of covariates; the variance decreases rapidly with the sample size. We see a similar pattern for $\widehat{Q}(\widehat{\boldsymbol{\pi}}^*)$ in right panel of Fig. 4, where the bias is nominal with a small number of factors and the variance contributes more prominently to the overall RMSE, which still decreases with the sample size. Results are consistent with the idea that the optimal stochastic interventions are more difficult to estimate if there are more candidate features involved.

We next compare the value of the optimal stochastic intervention, $Q(\widehat{\boldsymbol{\pi}}^*)$, against a simple baseline policy that, for each factor, places all mass on the level with the highest estimated main effect from a main-effects-only model (i.e., a degenerate policy selecting per-factor AMCE maximizers, ignoring interactions). As shown in Figure 5, the optimal SI method yields a higher mean value of 1.550 compared to the baseline, demonstrating the benefits of accounting for interactions in policy optimization.

We next consider estimated uncertainties compared against true sampling uncertainties. We see in Fig. A.I.1 that the asymptotic variance of $\widehat{Q}(\widehat{\boldsymbol{\pi}}^*)$ is somewhat underestimated

---

[3] For simplicity, we here do not adopt the sum-to-0 coefficient constraint, and instead use a baseline category.

Table 1: Comparing different approaches to conjoint analysis. Pr here refers to the data-generating probability distribution over candidate features; $\Pr_{\boldsymbol{\pi}}$ refers to the distribution defining an optimal stochastic intervention. SI denotes "stochastic intervention"; GLM denotes "generalized linear model."

| | *Average Marginal Component Effect (AMCE)* | *Average Marginal Interaction Effect (AMIE)* | *Average Case Optimal Stochastic Intervention* | *Adversarial Case Optimal Stochastic Intervention* |
|---|---|---|---|---|
| *Character* | | | | |
| Components considered at a time | 1 | 2+ | All | All |
| Baseline factor category specified? | Yes | Yes | No | No |
| Marginalization over: | Respondents; other factors of reference profile via Pr; all factors of opponent profile via Pr | Respondents; other factors of reference profile via Pr; all factors of opponent profile via Pr | Respondents; factors of reference via $\Pr_{\boldsymbol{\pi}}$, opponent profile via Pr | Respondents; factors of reference profile via $\Pr_{\boldsymbol{\pi}^a}$, opponent profile via $\Pr_{\boldsymbol{\pi}^b}$ |
| Informative about strategy in an adversarial setting? | No | No | No | Yes |
| Hyper-parameters | Strength of regularization in outcome model (rarely used) | Strength of regularization in outcome model if used | Strength of regularization in outcome model; SI regularization | Strength of regularization in outcome model; SI regularization |
| Uncertainty estimation | GLM variance-covariance; bootstrap | GLM variance-covariance; bootstrap | GLM variance-covariance + Delta method | GLM variance-covariance + Delta method |
| *Data Requirements* | | | | |
| Requires forced-choice design? | No | No | No | Yes |
| Requires distinct respondent and profile sub-groups? | No | No | No | Yes |

for small sample sizes. Fig. A.I.2 in §A.I.8 reports the true sampling variability of $\widehat{\boldsymbol{\pi}}^*$ against the average standard error estimate from asymptotic inference; estimates are neither systematically too wide nor too narrow. Finally, we examine coverage, which combines information about point with variance estimates. We see in Fig. 8 coverage close to the target coverage rate across the number of factors and observations for the components of $\boldsymbol{\pi}^*$ and (in Fig. 9) for $\widehat{Q}(\widehat{\boldsymbol{\pi}}^*)$ itself.

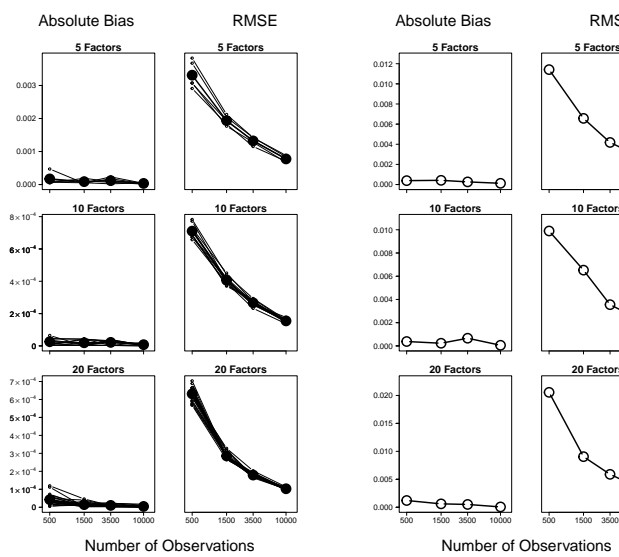

Figure 4: LEFT. Estimation bias and RMSE of $\boldsymbol{\pi}^*$. Each line represents one entry in $\boldsymbol{\pi}^*$. The bold line and closed circles represent the average value.
RIGHT. The estimation bias and RMSE of $Q(\boldsymbol{\pi}^*)$.

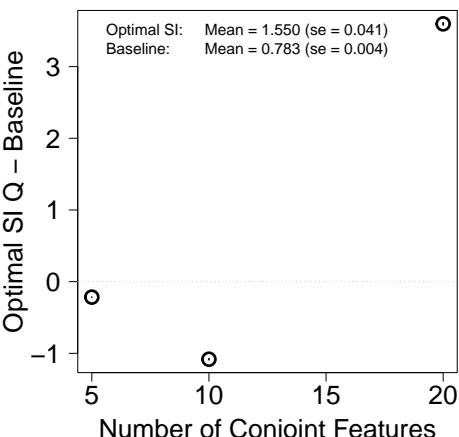

Figure 5: Comparison against a baseline.

### A.4 SIMULATION DESIGN: ADVERSARIAL CASE

In order to investigate finite-sample performance under the more complex adversarial setting, we simulate strategic behavior between two hypothetical political parties denoted by $R$ (Republican) and $D$ (Democrat). We design a two-stage electoral process in which each party first selects a nominee via a primary election, and then those nominees compete in a general election. Voters differ by party affiliation, which determines whether they participate in the corresponding primary. Let $p_R$ denote the fraction of Republican voters in the electorate, so that $p_D = 1 - p_R$ is the fraction of Democratic voters. For each simulation run, we fix $p_R \in \{0.2, 0.3, 0.5, 0.65, 0.8\}$ along a grid, and we vary the conjoint sample size $n \in \{1000, 5000, 10000\}$. Each grid cell is replicated across Monte Carlo draws.

Within each simulated dataset, we generate responses for primary and general-election stages. In the first stage, only voters from party $R$ or party $D$ participate in their own party's primary. We assign two potential candidate profiles for party $R$ and two for party $D$; one of these candidates is selected using the party's assignment mechanism, the other uniformly. Let these be $\mathbf{T}_i^{R,1}, \mathbf{T}_i^{R,2}$ for $R$ and $\mathbf{T}_i^{D,1}, \mathbf{T}_i^{D,2}$ for $D$. We specify probabilities with which

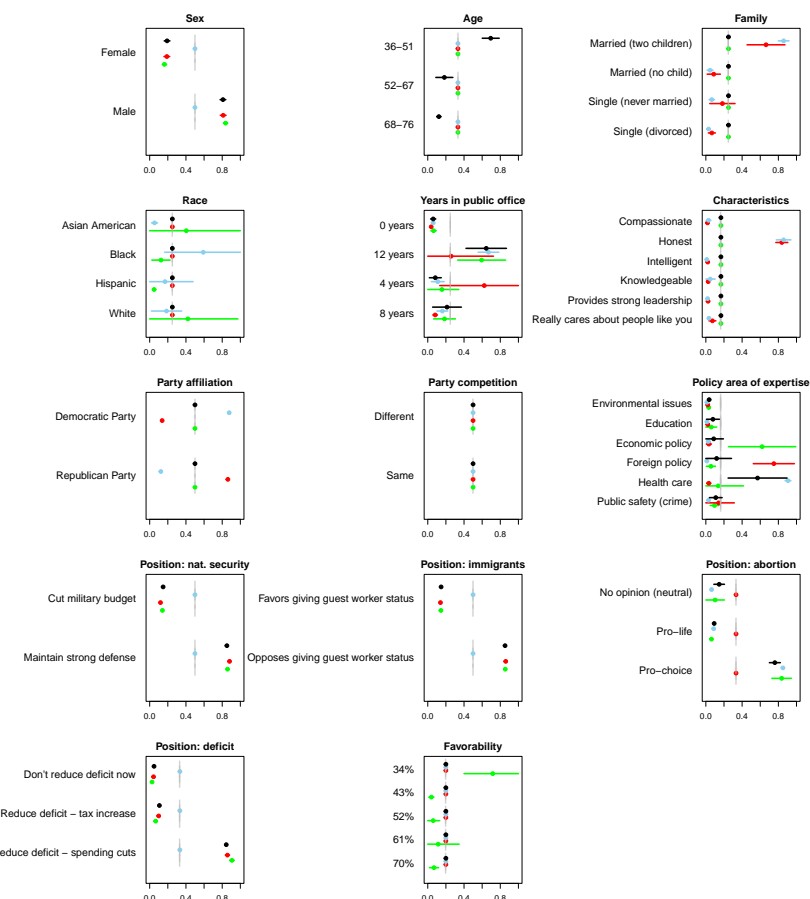

Figure 6: Optimal strategies in the average case setting. Black, blue, red, and green denote the average case optimal among all, Democrat, Republican, and Independent respondents in the sample.

each candidate profile is chosen by each respondent in that primary, using logistic models to capture how voters respond to candidate features—here, simply gender for tractability when computing ground truth equilibria via grid search. In the second stage, *all* voters, $R$ and $D$, face a forced choice in the general election between $\mathbf{T}_i^{R,*}$ and $\mathbf{T}_i^{D,*}$, winners of the respective primaries. In the second stage, Republican and Democrat voters select candidates again using two logistic models.

Having outlined the data-generating process, we now discuss how we compute the ground-truth strategies approximating a Nash equilibrium in the space of possible profile distributions for each party. The quantities $\boldsymbol{\pi}^R$ and $\boldsymbol{\pi}^D$ describes a mixed strategy over candidate characteristics for $R$ and $D$, respectively. We define

$$Q\big(\boldsymbol{\pi}^R, \boldsymbol{\pi}^D\big) \;=\; \mathbb{E}_{\mathbf{T}_i^R \sim \boldsymbol{\pi}^R,\, \mathbf{T}_i^D \sim \boldsymbol{\pi}^D}\Big[\Pr\big(Y_i(\mathbf{T}_i^R) > Y_i(\mathbf{T}_i^D)\big)\Big],$$

where $\mathbf{T}_i^R$ and $\mathbf{T}_i^D$ represent each party's selection (who competes against the primary challenger). To find a Nash equilibrium, we compute each party's best response to the other via a discrete grid search. In practice, this means we scan over $\boldsymbol{\pi}^R$ and $\boldsymbol{\pi}^D$, computing

$$\max_{\boldsymbol{\pi}^R} \min_{\boldsymbol{\pi}^D} \; Q\big(\boldsymbol{\pi}^R, \boldsymbol{\pi}^D\big) \;=\; \min_{\boldsymbol{\pi}^D} \max_{\boldsymbol{\pi}^R} \; Q\big(\boldsymbol{\pi}^R, \boldsymbol{\pi}^D\big),$$

checking which $(\boldsymbol{\pi}^R, \boldsymbol{\pi}^D)$ satisfies the equilibrium condition that neither party can unilaterally improve its expected vote share; we label these as the equilibrium strategies.

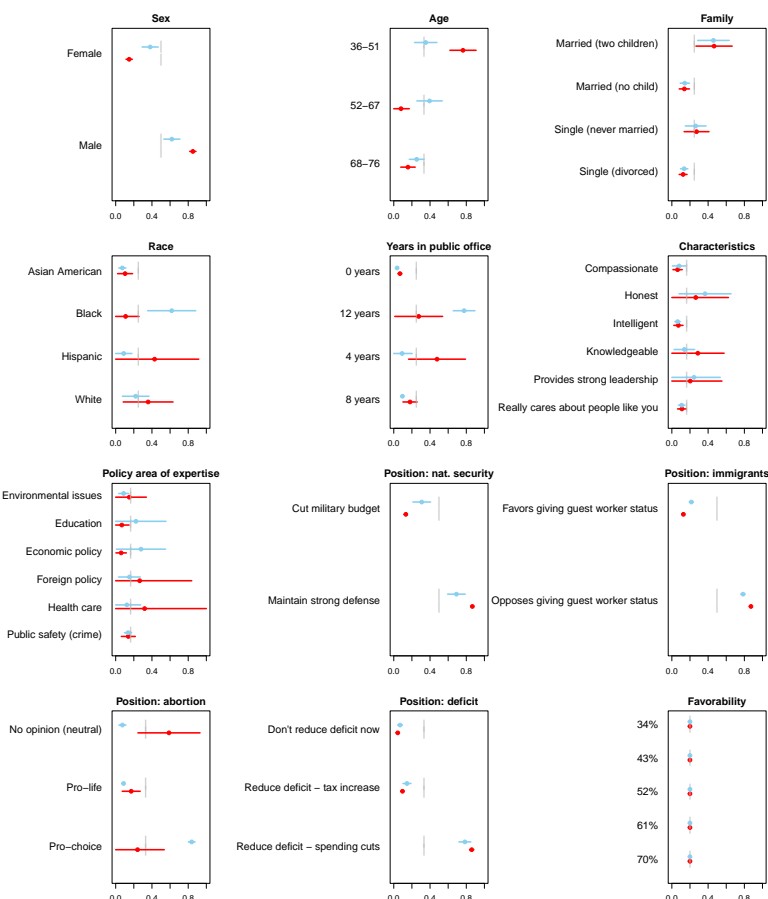

Figure 7: Optimal strategies in an adversarial setting. Blue/red denote the equilibrium strategy for the agent facing Democratic/Republican voters in the primary stage, respectively.

In order to evaluate the finite-sample performance of the proposed algorithm for the adversarial setting, we implemented the two-stage design described in the preceding section while varying the proportion of Republican voters, $p_R$, in the electorate. That is, for each Monte Carlo run, we save the estimated equilibrium distribution for $\pi^R$ and $\pi^D$, along with the realized general-election outcomes under those strategies. By aggregating results across the grid of $\{p_R, n_{\text{obs}}\}$ and across replications, we examine trends in how party composition $p_R$ and sample size $n_{\text{obs}}$ affect equilibrium strategies, estimated vote shares, and convergence. This design allows us to evaluate the proposed adversarial methodology under changing population compositions and sample sizes. We focus on summarizing estimation accuracy for $\pi^R$: we record root-mean-squared error (RMSE) and coverage of confidence intervals under repeated sampling, with coverage targeting the nominal rate of 95%.

## A.5 Empirical Results Referenced in Main Text

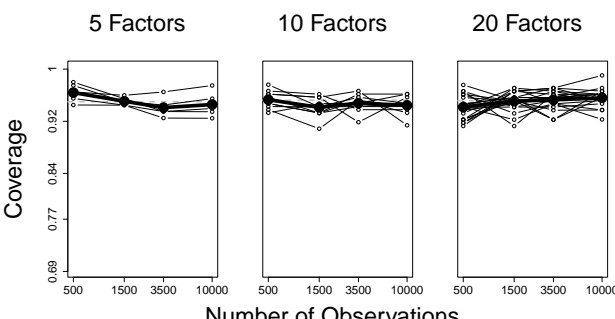

Figure 8: Finite-sample coverage of $\boldsymbol{\pi}^*$. Each line represents one entry in $\boldsymbol{\pi}^*$. The bold line and closed circles represent the average value.

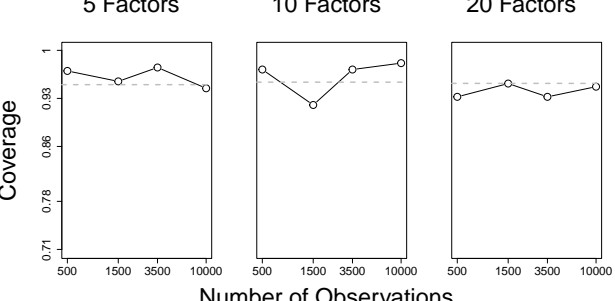

Figure 9: Finite-sample coverage for $Q(\boldsymbol{\pi}^*)$.

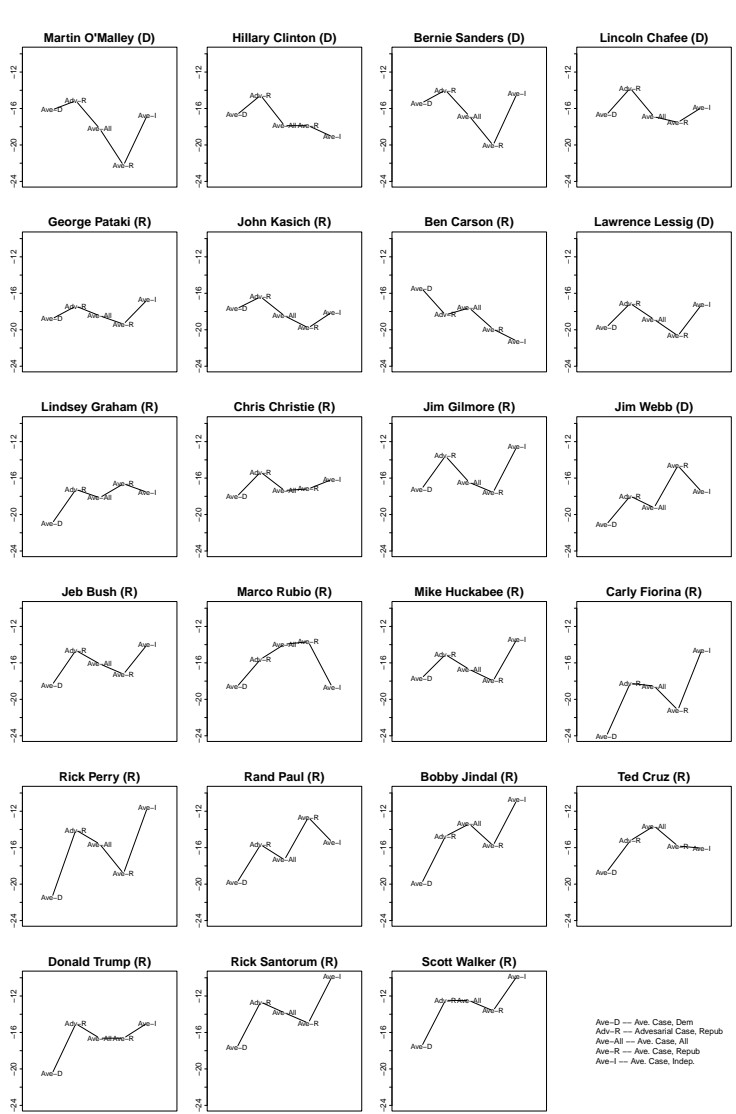

Figure 10: Candidate analysis. In all plots, the $y$ axis displays the evaluated log probability under a given policy. Figure sorted from left to right, arranged by how strongly their log probabilities trend across partisan ordering.

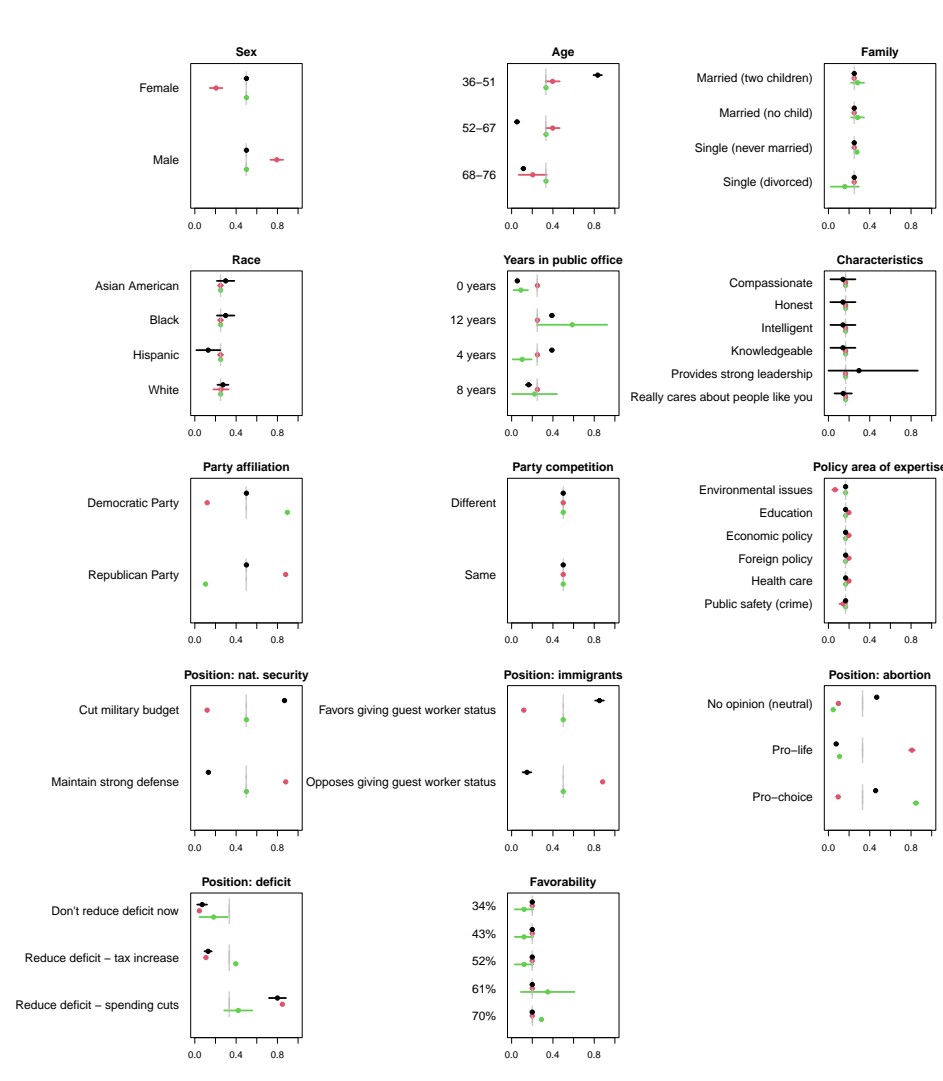

Figure 11: Optimal strategies in the covariate sensitive case, where a different strategy for allocating candidate features can be used for three data-derived clusters of voters.

# Supplementary Appendices

## Appendix I: Theoretical Analysis

### A.I.1 The Optimal Stochastic Intervention in a Two-Way Interaction Model With Binary Factors

Recall that the objective function to maximize is

$$O(\boldsymbol{\pi}) = Q(\boldsymbol{\pi}) - \lambda ||\mathbf{p} - \boldsymbol{\pi}||^2$$

$$= \beta_0 + \sum_{d=1}^{D} \beta_d \pi_d + \sum_{d' < d''} \gamma_{d',d'} \pi_{d'} \pi_{d''} - \lambda \sum_{d'''=1}^{D} \{(\pi_{d'''} - p_{d'''})^2 + ([1 - \pi_{d'''}] - [1 - p_{d'''}])^2\}$$

so that

$$\frac{dO}{d\pi_d} = \beta_d + \sum_{d,d'' \neq d'} \gamma_{d,d'} \pi_d' - 4\lambda(\pi_d - p_d) = 0$$

$$\implies$$

$$\sum_{d,d' \neq d} \gamma_{d,d'} \pi_{d''} - 4\lambda \pi_d = -\beta_d - 4\lambda p_d$$

where we use $\mathbf{p}_d$ to denote the vector of Categorical probabilities for all levels in factor $d$. This sets up a system of $D$ linear equations with $D$ unknowns, which can be represented in matrix form:

$$\mathbf{C}\boldsymbol{\pi}^* = \mathbf{B}$$
$$\boldsymbol{\pi}^* = \mathbf{C}^{-1}\mathbf{B},$$

where $B_{d,1} = -\beta_d - 4\lambda p_d$, $C_{d,d} = -4\lambda$ and $C_{d,d''} = \gamma_{d,d'}$.

### A.I.2 The Optimal Stochastic Intervention in a Two-Way Interaction Model With Multiple Factor Levels

The outcome model with multiple factor levels is

$$Y_i(t) = \beta_0 + \sum_{d=1}^{D} \sum_{l=1}^{L_d-1} \beta_{dl} t_{dl} + \sum_{d',d'':d'<d''} \sum_{l'=1}^{L_{d'}-1} \sum_{l''=1}^{L_{d''}-1} \gamma_{d'l',d''l''}\ t_{d'l'}\ t_{d''l''} + \epsilon_i,$$

where $t_{dl}$ denotes the binary indicator for whether level $l$ in factor $d$ is assigned. By linearity of expectations and independence of factors:

$$Q(\boldsymbol{\pi}) = \beta_0 + \sum_{d=1}^{D} \sum_{l=1}^{L_d-1} \beta_{dl} \pi_{dl} + \sum_{d',d'':d'<d''} \sum_{l'=1}^{L_{d'}-1} \sum_{l''=1}^{L_{d''}-1} \gamma_{d'l',d''l''}\ \pi_{d'l'}\ \pi_{d''l''}.$$

The objective is now

$$O(\boldsymbol{\pi}) = Q(\boldsymbol{\pi}) - \lambda ||\mathbf{p} - \boldsymbol{\pi}||^2$$

$$= \beta_0 + \sum_{d=1}^{D} \sum_{l=1}^{L_d-1} \beta_{dl}\ \pi_{dl} + \sum_{d',d'':d'<d''} \sum_{l'=1}^{L_{d'}-1} \sum_{l''=1}^{L_{d''}-1} \gamma_{d'l',d''l''}\ \pi_{d'l'}\pi_{d''l''}$$

$$- \lambda \sum_{d'''=1}^{D} \left\{ \sum_{l'''=1}^{L_{d'''}-1} (\pi_{d'''l'''} - p_{d'''l'''})^2 + \left(1 - \left[\sum_{l''''=1}^{L_{d'''}-1} \pi_{d'''l''''}\right] - \left(1 - \left[\sum_{l''''=1}^{L_{d'''}-1} p_{d'''l''''}\right]\right)\right)^2 \right\}$$

so that, for $l < L_d$:

$$\frac{dO}{d\pi_{dl}} = \beta_{dl} + \sum_{d,d' \neq d} \sum_{l=1}^{L_d-1} \sum_{l'=1}^{L_{d'}-1} \gamma_{dl,d'l'} \ \pi_{d'l'} - 2\lambda(\pi_{dl} - p_{dl}) - 2\lambda \left( \sum_{l'=1}^{L_d-1} (\pi_{dl'} - p_{dl'}) \right) = 0$$

$$\implies$$

$$\sum_{d,d' \neq d} \sum_{l=1}^{L_d-1} \sum_{l'=1}^{L_{d'}-1} \gamma_{dl,d'l'} \ \pi_{d'l'} - 4\lambda\pi_{dl} - 2\lambda \sum_{l' \neq l, l' < L_d} \pi_{dl'} = -\beta_{dl} - 4\lambda p_{dl} - 2\lambda \sum_{l' \neq l, l' < L_d} p_{dl'}$$

with This again sets up a system of $\sum_{d=1}^{D}(L_d - 1)$ linear equations with the same number of unknowns, which can be represented in matrix form:

$$\mathbf{C}\boldsymbol{\pi}^* = \mathbf{B}$$

$$\boldsymbol{\pi}^* = \mathbf{C}^{-1}\mathbf{B}.$$

where, letting $r(\cdot)$ denote a function returning the appropriate index into the matrix rows/columns:

$$B_{r(dl),1} = -\beta_{dl} - 4\lambda p_{dl} - 2\lambda \sum_{l' \neq l, l' < L_d} p_{dl'}$$

$$C_{r(dl),r(dl)} = -4\lambda$$

$$C_{r(dl),r(dl')} = -2\lambda$$

$$C_{r(dl),r(d'l'')} = \gamma_{dl,d'l''}$$

### A.I.3  The Optimal Stochastic Intervention in a Two-Way Interaction Model Under Forced Choice Outcomes

The outcome model with multiple factor levels is

$$\Pr(Y_i(\mathbf{T}_i^a) > Y_i(\mathbf{T}_i^b) \mid \mathbf{T}_i^a, \mathbf{T}_i^b) = \tilde{\mu} + \sum_{d=1}^{D} \sum_{l=1}^{L_d} \beta_{dl} \left( \mathbb{I}\{T_{id}^a = l\} - \mathbb{I}\{T_{id}^b = l\} \right)$$

$$+ \sum_{d',d'':d' < d''} \sum_{l'=1}^{L_{d'}} \sum_{l''=1}^{L_{d''}} \gamma_{d'l',d''l''} \left( \mathbb{I}\{T_{id'}^a = l', T_{id''}^a = l''\} - \mathbb{I}\{T_{id'}^b = l', T_{id''}^b = l''\} \right) + \epsilon_i,$$

where $t_{dl}$ denotes the binary indicator for whether level $l$ in factor $d$ is assigned. By linearity of expectations and independence of factors:

$$Q(\boldsymbol{\pi}^a, \boldsymbol{\pi}^b) = \mathbb{E}_{\boldsymbol{\pi}^a(\mathbf{T}^a), \boldsymbol{\pi}^b(\mathbf{T}^b)} \left[ \Pr(Y_i(\mathbf{T}_i^a) > Y_i(\mathbf{T}_i^b) \mid \mathbf{T}_i^a, \mathbf{T}_i^b) \right]$$

$$= \tilde{\mu} + \sum_{d=1}^{D} \sum_{l=1}^{L_d} \beta_{dl} \left( \pi_{dl}^* - \pi_{dl}^b \right)$$

$$+ \sum_{d',d'':d' < d''} \sum_{l'=1}^{L_{d'}} \sum_{l''=1}^{L_{d''}} \gamma_{d'l',d''l''} \left( \pi_{d'l'}^* \pi_{d'l'}^* - \pi_{d'l'}^b \pi_{d'l'}^b \right)$$

- Case 0: Choose $\pi^a$ and $\boldsymbol{\pi}^b$ jointly to maximize the selection probability for $\mathbf{T}_i^a$. Problem: Not an interpretable solution: Choose best candidate strategy $A$ to go against worst possible candidate $B$.
- Case 1: Average Case Analysis: Set $\boldsymbol{\pi}^b$ to be $\mathbf{p}$. Interpretation: Best candidate strategy $A$ uniformly averaging over all possible candidate $B$'s.
- Case 2: Minimax Analysis: Set $\boldsymbol{\pi}^a$ to maximize, $\boldsymbol{\pi}^b$ to minimize objective. Interpretation: Optimally select candidate strategy $A$ to compete against optimally selected candidate strategy $B$.

The objective is now

$$O(\boldsymbol{\pi}^a, \boldsymbol{\pi}^b) = Q(\boldsymbol{\pi}^a, \boldsymbol{\pi}^b) - \lambda \left( ||\mathbf{p} - \boldsymbol{\pi}^a||^2 + ||\mathbf{p} - \boldsymbol{\pi}^b||^2 \right)$$

$$= \tilde{\mu} + \sum_{d=1}^{D} \sum_{l=1}^{L_d} \beta_{dl} \left( \pi_{dl}^* - \pi_{dl}^b \right)$$

$$+ \sum_{d',d'':d'<d''} \sum_{l'=1}^{L_{d'}} \sum_{l''=1}^{L_{d''}} \gamma_{d'l',d''l''} \left( \pi_{d'l'}^* \pi_{d'l'}^* - \pi_{d'l'}^b \pi_{d'l'}^b \right)$$

$$- \lambda \sum_{\# \in \{*,b\}} \sum_{d'''=1}^{D} \left\{ \sum_{l'''=1}^{L_d} (\pi_{d'''l'''}^{\#} - p_{d'''l'''})^2 \right\}$$

Under the Average Case Maximizer:

$$O(\boldsymbol{\pi}^a, \mathbf{p}) = \tilde{\mu} + \sum_{d=1}^{D} \sum_{l=1}^{L_d} \beta_{dl} \left( \pi_{dl}^* - p_{dl} \right)$$

$$+ \sum_{d',d'':d'<d''} \sum_{l'=1}^{L_{d'}} \sum_{l''=1}^{L_{d''}} \gamma_{d'l',d''l''} \left( \pi_{d'l'}^* \pi_{d'l''}^* - p_{d'l'} p_{d''l''} \right)$$

$$- \lambda \sum_{d'''=1}^{D} \left\{ \sum_{l'''=1}^{L_d} (\pi_{d'''l'''}^* - p_{d'''l'''})^2 \right\}$$

so that

$$\frac{dO}{d\pi_{dl}^*} = \beta_{dl} + \sum_{d,d' \neq d} \sum_{l=1}^{L_d} \sum_{l'=1}^{L_{d'}} \gamma_{dl,d'l'} \ \pi_{d'l'}^* - 2\lambda(\pi_{dl}^* - p_{dl}) = 0$$

$$\Longrightarrow$$

$$\sum_{d,d' \neq d} \sum_{l=1}^{L_d} \sum_{l'=1}^{L_{d'}} \gamma_{dl,d'l'} \ \pi_{d'l'}^* - 2\lambda \pi_{dl}^* = -\beta_{dl} - 2\lambda p_{dl}$$

This sets up a system of $\sum_{d=1}^{D} L_d$ linear equations with the same number of unknowns, which can be represented in matrix form:

$$\mathbf{C} \boldsymbol{\pi}^* = \mathbf{B}$$

$$\boldsymbol{\pi}^* = \mathbf{C}^{-1} \mathbf{B}.$$

where, letting $r(\cdot)$ denote a function returning the correct index into the matrix:

$$B_{r(dl),1} = -\beta_{dl} - 2\lambda p_{dl}$$

$$C_{r(dl),r(dl)} = -2\lambda$$

$$C_{r(dl),r(dl')} = 0$$

$$C_{r(dl),r(d'l'')} = \gamma_{dl,d'l''}$$

Here, the optimal stochastic intervention is a deterministic function of the outcome model parameters. The parameters defining the outcome model, $\boldsymbol{\beta}$ and $\boldsymbol{\gamma}$, are not known *a priori*, but can be estimated via generalized linear methods, with the asymptotic standard errors then employed. Because the parameters, $\boldsymbol{\pi}^a$, that define $\mathrm{Pr}_{\boldsymbol{\pi}}^a$ are a deterministic function of the regression parameters, the variance-covariance matrix of $\{\widehat{Q}(\widehat{\boldsymbol{\pi}}^a), \widehat{\boldsymbol{\pi}}^a\}$ can be obtained via the delta method:

$$\text{Var-Cov}(\{\widehat{Q}(\widehat{\boldsymbol{\pi}}^a), \widehat{\boldsymbol{\pi}}^a\}) = \mathbf{J} \ \hat{\Sigma} \ \mathbf{J}',$$

where $\hat{\Sigma}$ is the variance-covariance matrix from the modeling strategy for $Y_i$ using regression parameters $\boldsymbol{\beta}$ and $\boldsymbol{\gamma}$ and $\mathbf{J}$ is the Jacobian of partial derivatives (e.g., of $\widehat{Q}(\widehat{\boldsymbol{\pi}}^a)$ and $\widehat{\boldsymbol{\pi}}^a$ with respect to the outcome model parameters):

$$\mathbf{J} = \boldsymbol{\nabla}_{\{\hat{\boldsymbol{\beta}}, \hat{\boldsymbol{\gamma}}\}} \{\widehat{Q}(\widehat{\boldsymbol{\pi}}^{a^a}), \widehat{\boldsymbol{\pi}}^{a^a}\}.$$

If the assumptions of the first-stage model hold, then

$$\sqrt{n}\left(\{\widehat{Q}(\widehat{\boldsymbol{\pi}}^a), \widehat{\boldsymbol{\pi}}^a\} - \{Q(\boldsymbol{\pi}^a), \boldsymbol{\pi}^a\}\right) \to \mathcal{N}\left(\mathbf{0},\ \mathbf{J}\,\Sigma_n\,\mathbf{J}'\right)$$

### A.I.4 GRADIENTS FOR OBTAINING THE VARIANCE-CONSTRAINED OPTIMAL STOCHASTIC INTERVENTION IN THE TWO-WAY CONSTRAINED CASE

The gradients for the simplex-constrained objective function are, $l < L_d$,

$$
\frac{\partial O}{\partial a_{dl}} = \beta_{dl} A_{dl} + \sum_{l^{(a)} \neq l} \beta_{dl^{(a)}} \left( \frac{-\exp(a_{dl})\exp(a_{dl^{(a)}})}{\{1 + \sum_{l^{(m)}=1}^{L_d-1}\exp(a_{dl^{(m)}})\}^2} \right)
$$

$$
+ \sum_{d,d'\neq d} \sum_{l'=1}^{L_{d'}-1} \gamma_{dl,d'l'} A_{dl}\ \frac{\exp(a_{d'l'})}{\{1 + \sum_{l'^{(m)}=1}^{L_{d'}-1}\exp(a_{d'l'^{(m)}})\}}
$$

$$
+ \sum_{d,d'\neq d} \sum_{l^{(a)}\neq l} \sum_{l'=1}^{L_{d'}-1} \gamma_{dl^{(a)},d'l'}\ \frac{-\exp(a_{dl})\exp(a_{dl^{(a)}})}{\{1 + \sum_{l^{(m)}=1}^{L_d-1}\exp(a_{dl^{(m)}})\}^2}\ \frac{\exp(a_{d'l'})}{\{1 + \sum_{l'^{(m)}=1}^{L_{d'}-1}\exp(a_{d'l'^{(m)}})\}}
$$

$$
- 2\lambda A_{dl} \left( \frac{\exp(a_{dl})}{\{1 + \sum_{l^{(m)}=1}^{L_d-1}\exp(a_{dl^{(m)}})\}} - p_{dl} \right)
$$

$$
- 2\lambda \sum_{l^{(a)}\neq l} \left\{ \frac{-\exp(a_{dl})\exp(a_{dl^{(a)}})}{\{1 + \sum_{l^{(m)}=1}^{L_d-1}\exp(a_{dl^{(m)}})\}^2} \left( \frac{\exp(a_{dl^{(a)}})}{\{1 + \sum_{l^{(m)}=1}^{L_d-1}\exp(a_{dl^{(m)}})\}} - p_{dl^{(a)}} \right) \right\}
$$

$$
- 2\lambda \frac{\exp(a_{dl})}{\{1 + \sum_{l^{(m)}=1}^{L_d-1}\exp(a_{dl^{(m)}})\}^2} \sum_{l'=1}^{L_d-1} \left( \frac{\exp(a_{dl'})}{\{1 + \sum_{l'^{(m)}=1}^{L_d-1}\exp(a_{dl'^{(m)}})\}} - p_{dl'} \right),
$$

where

$$
A_{dl} = \frac{\exp(a_{dl})\{1 + \sum_{l^{(m)}=1}^{L_d-1}\exp(a_{dl^{(m)}}) - \exp(a_{dl})\}}{\{1 + \sum_{l^{(m)}=1}^{L_d-1}\exp(a_{dl^{(m)}})\}^2}.
$$

### A.I.5 GRADIENTS FOR OBTAINING THE VARIANCE-CONSTRAINED OPTIMAL STOCHASTIC INTERVENTION IN THE TWO-WAY CONSTRAINED CASE UNDER A FULLY PARAMETERIZED MODEL UNDER FORCED CHOICE

The objective is

$$
O(\boldsymbol{\pi}^a, \boldsymbol{\pi}^b) = Q(\boldsymbol{\pi}^a, \boldsymbol{\pi}^b) - \lambda\left(||\mathbf{p} - \boldsymbol{\pi}^a||^2 + ||\mathbf{p} - \boldsymbol{\pi}^b||^2\right)
$$

$$
= \tilde{\mu} + \sum_{d=1}^{D}\sum_{l=1}^{L_d} \beta_{dl}\left(\pi_{dl}^* - \pi_{dl}^b\right)
$$

$$
+ \sum_{d',d'':d'<d''}\sum_{l'=1}^{L_{d'}}\sum_{l''=1}^{L_{d''}} \gamma_{d'l',d''l''}\left(\pi_{d'l'}^*\pi_{d'l'}^* - \pi_{d'l'}^b\pi_{d'l'}^b\right)
$$

$$
- \lambda \sum_{\#\in\{*,b\}}\sum_{d'''=1}^{D} \left\{ \sum_{l'''=1}^{L_d}(\pi_{d'''l'''}^\# - p_{d'''l'''})^2 \right\}
$$

### A.I.6 Gradients for Obtaining the Variance-Constrained Optimal Stochastic Intervention in the Two-Way Constrained Case under a Fully Parameterized Model

With a fully parameterized, ANOVA-type model, we have:

$$Q(\mathbf{a}) = \beta_0 + \sum_{d=1}^{D} \sum_{l=1}^{L_d} \beta_{dl} \cdot \frac{1}{1 + \exp(-a_{dl})} + \sum_{d',d'':d'<d''} \sum_{l'=1}^{L_{d'}} \sum_{l''=1}^{L_{d''}} \gamma_{d'l',d''l''} \ \pi_{d'l'} \pi_{d''l''}.$$

The gradients for the simplex-constrained objective function are, $l < L_d$,

$$\frac{\partial O}{\partial a_{dl}} = \beta_{dl} A_{dl} + \sum_{l^{(a)} \neq l} \beta_{dl^{(a)}} \left( \frac{-\exp(a_{dl})\exp(a_{dl^{(a)}})}{\{1 + \sum_{l^{(m)}=1}^{L_d-1} \exp(a_{dl^{(m)}})\}^2} \right)$$

$$+ \sum_{d,d' \neq d} \sum_{l'=1}^{L_{d'}} \gamma_{dl,d'l'} A_{dl} \ \frac{\exp(a_{d'l'})}{\{1 + \sum_{l'^{(m)}=1}^{L_{d'}-1} \exp(a_{d'l'^{(m)}})\}}$$

$$+ \sum_{d,d' \neq d} \sum_{l^{(a)} \neq l} \sum_{l'=1}^{L_{d'}} \gamma_{dl^{(a)},d'l'} \ \frac{-\exp(a_{dl})\exp(a_{dl^{(a)}})}{\{1 + \sum_{l^{(m)}=1}^{L_d-1} \exp(a_{dl^{(m)}})\}^2} \ \frac{\exp(a_{d'l'})}{\{1 + \sum_{l'^{(m)}=1}^{L_{d'}-1} \exp(a_{d'l'^{(m)}})\}}$$

$$- 2\lambda A_{dl} \left( \frac{\exp(a_{dl})}{\{1 + \sum_{l^{(m)}=1}^{L_d-1} \exp(a_{dl^{(m)}})\}} - p_{dl} \right)$$

$$- 2\lambda \sum_{l^{(a)} \neq l} \left\{ \frac{-\exp(a_{dl})\exp(a_{dl^{(a)}})}{\{1 + \sum_{l^{(m)}=1}^{L_d-1} \exp(a_{dl^{(m)}})\}^2} \left( \frac{\exp(a_{dl^{(a)}})}{\{1 + \sum_{l^{(m)}=1}^{L_d-1} \exp(a_{dl^{(m)}})\}} - p_{dl^{(a)}} \right) \right\}$$

$$- 2\lambda \frac{\exp(a_{dl})}{\{1 + \sum_{l^{(m)}=1}^{L_d-1} \exp(a_{dl^{(m)}})\}^2} \sum_{l'=1}^{L_d-1} \left( \frac{\exp(a_{dl'})}{\{1 + \sum_{l'^{(m)}=1}^{L_d-1} \exp(a_{dl'^{(m)}})\}} - p_{dl'} \right),$$

where

$$A_{dl} = \frac{\exp(a_{dl})\{1 + \sum_{l^{(m)}=1}^{L_d-1} \exp(a_{dl^{(m)}}) - \exp(a_{dl})\}}{\{1 + \sum_{l^{(m)}=1}^{L_d-1} \exp(a_{dl^{(m)}})\}^2}.$$

### A.I.7 Objective Function in Unconstrained Space

$$O(\boldsymbol{a}) = Q(\boldsymbol{a}) - \lambda_n ||\mathbf{p} - \boldsymbol{\pi}||^2$$

$$= \beta_0 + \sum_{d=1}^{D} \sum_{l=1}^{L_d-1} \beta_{dl} \ \frac{\exp(a_{dl})}{\{1 + \sum_{l^{(m)}=1}^{L_d-1} \exp(a_{dl^{(m)}})\}}$$

$$+ \sum_{d',d'':d'<d''} \sum_{l'=1}^{L_{d'}-1} \sum_{l''=1}^{L_{d''}-1} \gamma_{d'l',d''l'''} \ \frac{\exp(a_{d'l'})}{\{1 + \sum_{l'^{(m)}=1}^{L_{d'}-1} \exp(a_{d'l'^{(m)}})\}} \frac{\exp(a_{d''l'})}{\{1 + \sum_{l'^{(m)}=1}^{L_{d''}-1} \exp(a_{d''l'^{(m)}})\}}$$

$$- \lambda_n \sum_{d'''=1}^{D} \left\{ \sum_{l'''=1}^{L_d-1} \left( \frac{\exp(a_{d'''l'''})}{\{1 + \sum_{l'''^{(m)}=1}^{L_{d'''}-1} \exp(a_{d'''l'''^{(m)}})\}} - p_{d'''l'''} \right)^2 \right.$$

$$\left. + \left( 1 - \left[ \sum_{l''''=1}^{L_{d'''}-1} \frac{\exp(a_{d'''l''''})}{\{1 + \sum_{l''''^{(m)}=1}^{L_{d'''}-1} \exp(a_{d'''l''''^{(m)}})\}} \right] - \left( 1 - \left[ \sum_{l''''=1}^{L_{d'''}-1} p_{d'''l''''} \right] \right) \right)^2 \right\}$$

## Appendix II: Supplementary Simulation Results

### A.I.8 Supplementary Simulation Results with the Two-Step Estimator

### A.I.9 Estimation Details

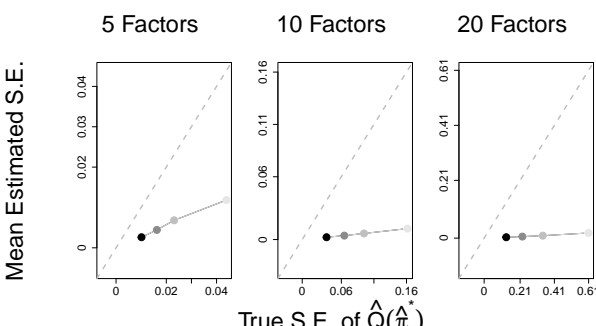

Figure A.I.1: Points depict the average estimated standard deviation obtained via the Delta method. Colors depict the sample size (with $n = 500$ being light gray and $n=10,000$ being black).

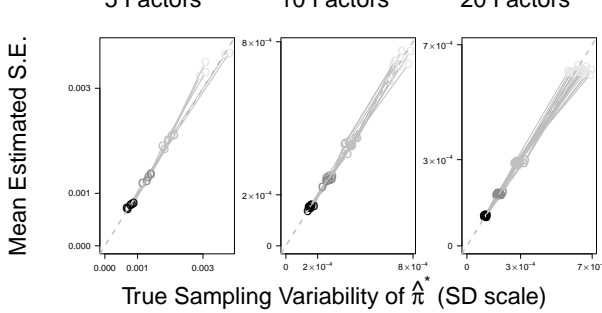

Figure A.I.2: True sampling variability of $\hat{\boldsymbol{\pi}}^*$ plotted against the variability estimated via asymptotic inference. Colors depict the sample size (with $n=500$ being light gray and $n=10,000$ being black).

# Appendix III: Additional Application Results

## A.III.1 Mapping the 2016 Candidate Primary Features onto the Conjoint Levels of Ono & Burden (2019)

We map the features of the 2016 presidential election candidates onto the conjoint features of Ono & Burden (2019). In some cases, this mapping is straightforward (e.g., with candidate gender). In other cases, the mapping is less straightforward. For example, the factor levels associated with marital status do not encompass the full range of possibilities seen among 2016 candidates. In such cases, we select the closest mapping (see Replication Data for full details). For example, a real, married candidate with 4 children would be mapped to the "Married with 2 children" level (not the "Single, divorced" or"Married, no children" levels).

We will explore these substantive questions by integrating the experiment mentioned above from Ono & Burden (2019). In this election, 17 Republican and 6 Democrat candidates vied for their respective partie' nomination in primaries. These candidates have a large number of features, which we mapped onto the conjoint factors of Ono & Burden (2019) (see §A.III.1 for details). Below we present this mapping for four of the candidates:

* *Ben Carson:* Republican, Black, male, 68-76, married (with children), 0 years of political experience, compassionate, policy focus on health care, emphasis on maintaining strong defense, opposes giving guest worker status, pro-life, don't reduce deficit now.

* *Hillary Clinton:* Democrat, White, female, 68-76, married (with children), 16 years of political experience, provides strong leadership, foreign policy, maintains strong defense, favors giving guest worker status, pro-choice, don't reduce deficit now.

* *Bernie Sanders:* Democrat, White, male, 68-76, married (with children), 34 years of political experience, compassionate, policy focus on economy, cut military budget, ambiguous position on immigration, pro-choice, reduce deficit through tax increase.

* *Donald Trump:* Republican, White, male, 68-76, married (with children), 0 years of political experience, provides strong leadership, policy focus on economy, emphasis on maintaining strong defense, opposes giving guest worker status, pro-life, reduce deficit through spending cuts.

*Implementation Details:* Implementation deploys JAX for performing differentiable optimization routines. All analyses run in under 12 hours on consumer-grade PC hardware (CPU or GPU).

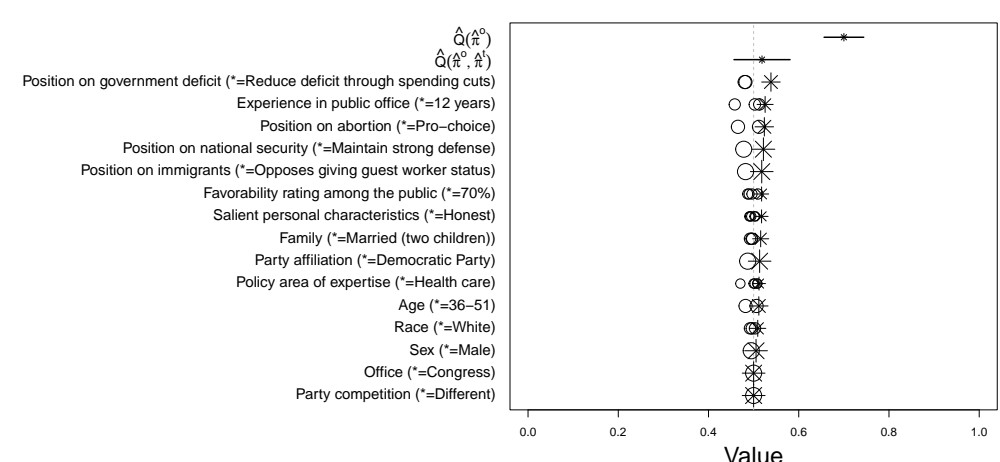

Figure A.III.1: Expected optimized vote share in the population in the average (uniform) case (denoted by $\widehat{Q}(\widehat{\boldsymbol{\pi}}^a)$), compared against factor-wise marginal means. The $*$ for the marginal means indicates the level with the highest marginal outcome (with that level listed on the right-hand side of the figure along with the factor name). $\widehat{Q}(\widehat{\boldsymbol{\pi}}^a, \widehat{\boldsymbol{\pi}}^b)$ denotes the expected optimized vote share in the population under adversarial conditions compared against factor-wise marginal means.

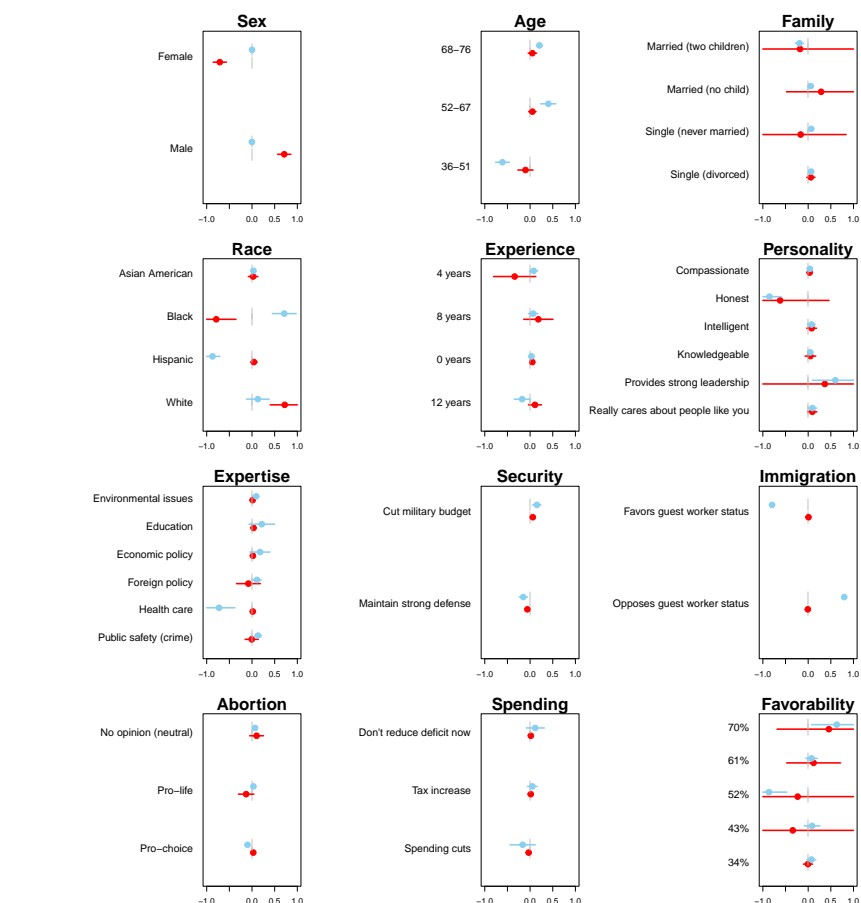

Figure A.III.2: Comparing optimal strategies in the non-adversarial vs. adversarial setting. Blue and red denote the equilibrium strategy for the agent facing Democratic and Republican voters in the primary stage, respectively.

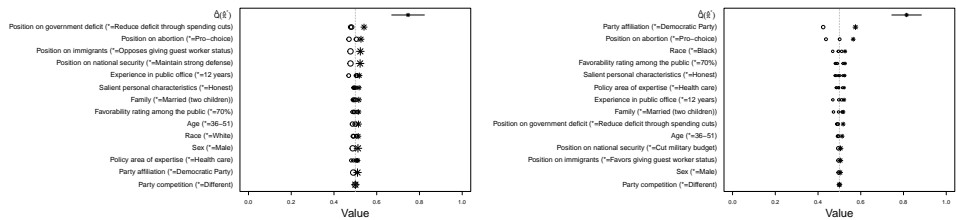

Figure A.III.3: Marginal means analysis, among all (left) and Democrat respondents (right).

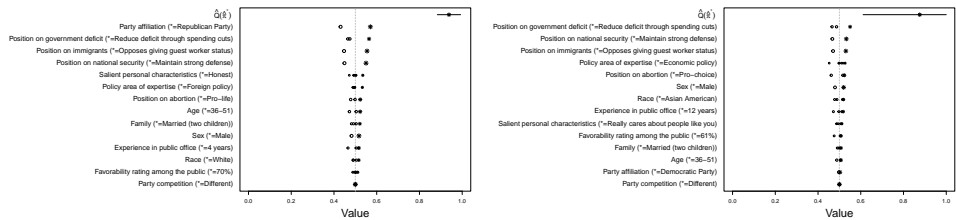

Figure A.III.4: Marginal means analysis, among Republican (left) and Independent (right)respondents.

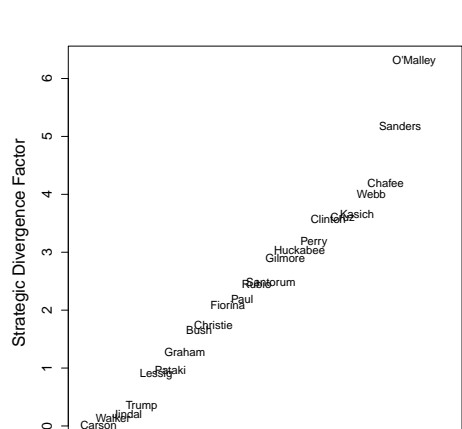

Figure A.III.5: Strategic divergence factor computed for major candidates in the 2016 primaries.

