# OpenReview forum: "MiniMax Learning of Interpretable Factored Stochastic Policies from Conjoint Data, with Uncertainty Quantification"
_ICLR.cc/2026/Conference — ICLR 2026 Conference Withdrawn Submission_

### Official Review · Reviewer_Wo8e · 2025-10-27

**Soundness:** 3
**Presentation:** 2
**Contribution:** 3
**Rating:** 4
**Confidence:** 2

**Summary:**

The authors address the problem of identifying optimal interventions when determining the importance of different features in the selection of political candidates. Specifically, the authors consider a setting with two candidates, 𝐴 and B, and multiple treatments or factor levels, and propose a framework for understanding the effects of these treatments as a policy optimization problem rather than through standard average treatment effect estimation.

The topic is theoritically and practically interesting, and the paper makes strong theoretical contributions. The formulation as a policy maximization problem provides a clear bridge between causal inference and decision-making, and the empirical evaluation is thorough and carefully designed.


Overall, this is an interesting and theoretically substantial paper, but it would benefit significantly from improved exposition—specifically, brief illustrative examples, restated notation reminders, and clearer intuition behind key quantities and theorems.

**Strengths:**

- Interesting topic
- Proposes a new method to adress the problem
- Clear applications
- Theoritical foundations

**Weaknesses:**

I found the paper difficult to follow at several points, primarily due to dense notation and limited intuitive explanation. The theoretical sections are rigorous but could benefit from more narrative guidance for the reader. For instance, the repeated references to “levels” would be clearer if accompanied by a simple, concrete example (e.g., levels corresponding to candidate attributes such as gender or policy stance). Similarly, after Equation (1), in the “forced-choice” conjoint formulation, the term Pr_p
is introduced without explicit definition; it would help to clarify that this denotes the known assignment (logging) distribution from the conjoint design, typically uniform unless otherwise specified.

**Questions:**

- Why it is better to leverage policy optimization over causal estimation beyond interpetability?
- Can you make a clear comparison on the quality of the optimal solutions you get vs classical approaches?

---

### Official Review · Reviewer_SZjm · 2025-10-31

**Soundness:** 3
**Presentation:** 1
**Contribution:** 3
**Rating:** 4
**Confidence:** 2

**Summary:**

This paper introduces a new framework for learning factored stochastic policies from conjoint experiment data, moving beyond traditional AMCE estimation. The authors model both average-case and adversarial (two-party) scenarios, allowing strategic interactions between candidates to be studied more realistically. Overall, I don’t have much background in this specific problem setup, but I find it somewhat difficult to understand the exact position of this paper in the current literature. Some necessary related work discussion is missing in the current version, which makes it hard to justify the paper’s contribution and positioning.

**Strengths:**

1. The paper is motivated by real data and is comprehensively studied using U.S. presidential conjoint data. I appreciate the detailed background introduction and problem description.
2. The theoretical derivation is rigorous, with a clear progression from the adversarial to the non-adversarial case.
3. Conceptually, I appreciate that the authors reframe conjoint analysis from effect estimation to policy optimization, which enables richer strategic insights.

**Weaknesses:**

1. The paper format does not strictly follow the ICLR template (the font type looks different from standard papers). The overall structure and layout also need refinement. For example, in Lines 420–427 and 467–478, the table is too small, and the figure captions are misplaced. Some equations are overfull and need adjustment, and the spacing between paragraphs is uneven. The paper should be carefully polished to meet publication standards.
2. In Figure 2, for the table in the middle, it’s unclear how the likelihood ratio is calculated (it’s not mentioned in the main text), and the standard error is also missing.
3. The paper lacks a sufficient related work section to summarize existing studies and developments. This makes it difficult to evaluate the novelty and contribution of the work.
4. Following point 3, comparisons with other methods from the literature in the simulation section would further strengthen the validation and show the advantage of the proposed method.
5. It is not very clear how generalizable the parametric model described in Equation (2) is beyond the political conjoint context. Including more motivating examples from other domains would help assess the broader applicability of the framework.

**Questions:**

See Weaknesses.

---

### Official Review · Reviewer_jpMP · 2025-11-01

**Soundness:** 2
**Presentation:** 1
**Contribution:** 3
**Rating:** 4
**Confidence:** 2

**Summary:**

This paper provides a framework of conjoint analysis and policy learning for both adversarial, and non-adversarial environments. It models the strategic interaction by min-max game, also it provides a closed-form solution and its uncertainty of the average-case optimizer for linear outcome model with two-way interactions. The paper provides both a synthetic data and U.S. presidential conjoint analysis to show the proposed method.

**Strengths:**

- The paprt provides an interpretable, factored stochastic policy learning with minimax optimization formulation, providing both a closed-form solution and a gradient-based solver makes the approach theoretically elegant and practically feasible.
- The Delta-method uncertainty quantification offers a statistically principled way to estimate confidence intervals on policy outcomes.
- The first work to address optimal profile selection, particularly in adversarial settings.

**Weaknesses:**

- The writing of the paper may need improvement. May use the default style, \section and \subsection to improve readability, also font size in figures and tables need to be enlarged, it's a little hard to follow the main methodology description. The paper is not well organized in the main body part, heavily rely on appendix.
- The experiemental section is mostly illustrative, and use case are not fully shown across domains. The theoretical derivations, especially around Eq. (4)–(7), are dense, with limited explanation of the underlying intuition/assumptions.
- The proposition and derivation do not hold for small $\lambda_n$, lacking closed form solution, but the paper has little discussion on practical use of iterative solution, as well as uncertainty estimations.

**Questions:**

- The paper mentioned high-dimensional cases with uncertainty quantification, could you elaborate more on the computational efficiency side, as it grows exponentially? It seems like that even for small dataset, it still took 12 hours.
- How can the method be generalized to the model to GLM/NN or >2-way interactions, as the paper claimed that can work for richer outcome models (including regularized GLMs and neural models)?
- Can you elaborate more on the baseline method and the comparison part, as being the
first to address optimal profile selection, what will be the difference/improvement?
- Does the iterative gradient-based solver for the adversarial case have any convergence guarantees (e.g., to a local Nash equilibrium)?
- How does performance change if we do not assume a uniform distribution for candidate?

---

### Official Review · Reviewer_cZoU · 2025-11-01

**Soundness:** 2
**Presentation:** 1
**Contribution:** 2
**Rating:** 4
**Confidence:** 2

**Summary:**

The paper considers the task of learning a policy from conjoint data.
They first consider the case of a 'static' environment, and then turn to a more involved adversarial/strategic scenario.
Besides deriving a fitting procedure, they also provide some uncertainty quantification by the means of asymptotic distributions of the relevant MLEs (derived via the Delta method).

**Strengths:**

First of all, I should say that my understanding of some topics central to this submission is quite limited, and so my review should be taken with a grain of salt.
In particular, I must say this is the first I've heard of conjoint analysis and conjoint data.

That said, the problem tackled in the paper -- conjoint analysis, and in particular the learning of interpretable, uncertainty-quantified policies from conjoint data -- is surely of relevance to a number of practitioners.

**Weaknesses:**

As mentioned, I am unfamiliar with much of the related literature.
Because of this, I am unable to assess the novelty of the paper's contributions.
It is also somewhat difficult for me to assess the broader impacts of this work; from what I gather, its impact essentially amounts to proposing a solution to the problem of learning policies from data arising from conjoint analysis.

In my opinion, the paper would greatly benefit from a brief overview of what conjoined analysis is, what problem it solves and how it is typically tackled.
The lack of such a section/paragraph makes the paper very hard to read for someone not already familiar with such concepts, and, as far as I am aware, these are not particularly popular to a machine learning audience.
Perhaps it would be helpful to contrast conjoint data analysis to more usual treatment effect estimation, which is decidedly more common in the ML literature and widely understood.

Also important to mention is that the current manuscript deviates substantially from the ICLR template.
In particular, there is no introduction heading, the headings that are present are all unnumbered and some grievous margin violations.
There are also numerous typography issues.
It is crucial that these issues be fixed before publication.

In terms of actual content, the derivations seemed reasonably sound, if a bit direct/simple (which is not necessarily a bad thing). I did, however, find some steps a bit sudden and not sufficiently motivated (e.g., in the transition from a static environment to an adversarial/strategic setting), but this may be just due to lack of context on my part.
I was unable to carefully check the math and computations.

Overall, my feeling is that the paper in its current state is not ready for publication at ICLR.
In particular, I do not believe that the current presentation is up to par to ICLR stanndards (even independent of the template/typesetting issues). It was a fairly hard read.
I encourage the authors to consider anchoring the paper to a topic that will be familiar to a wider part of the ICLR community, such as treatment effect estimation (which seems related, is more on the statistical inference side than the pure machine learning side and is widely known. Though other statistical topics might work too). In particular, I'd advise the authors to pay particular attention to the terminology being used (e.g., "candidate profiles") and to the motivation throughout the paper.

Altogether, these thoughts lead me to give a score of 'borderline reject'. \
**Why not higher:** The key reason is that I do not believe that the current presentation is up to par to ICLR standards. An additional factor is the seemingly limited broader impacts of the work. \
**Why not lower:** I am not confident enough in my rating to give a clear rejection, nor have I found anything in the paper to be unsound.

Again, it is possible that my issues with the paper are mainly due to a lack of context on my part. Should other reviewers have a more favorable impression of the paper, I would not mind its publication (assuming that more blatant things such as the formatting are resolved).

Remaining minor comments (+ some typography issues):
- Line 097: missing space after period
- Many paragraphs somehow start with a space. For example, on lines 125/126, 150/151, 156/157, 196/197, 201/202, and more.
- Some equations are grievously over the margin: lines 194-196 and 198-200
- The figures and tables are neigh illegible due to their sizes. This applies both to Figure 1 and Figure 2, but especially on Figure 2's table.

**Questions:**

- Could the authors give a brief overview of conjoint data analysis?
- Could the authors discuss the broader impacts of their work?

---

### Note · Authors · 2025-11-12

I have read and agree with the venue's withdrawal policy on behalf of myself and my co-authors.